# Global systematic review with meta-analysis shows that warming effects on terrestrial plant biomass allocation are influenced by precipitation and mycorrhizal association

Lingyan Zhou[1], Xuhui Zhou [1,2] ✉, Yanghui He[2], Yuling Fu[1], Zhenggang Du[2], Meng Lu[3], Xiaoying Sun[1], Chenghao Li[1], Chunyan Lu[1], Ruiqiang Liu[2], Guiyao Zhou [1], Shahla Hosseni Bai[4] & Madhav P. Thakur [5]

Biomass allocation in plants is fundamental for understanding and predicting terrestrial carbon storage. Yet, our knowledge regarding warming effects on root: shoot ratio (R/S) remains limited. Here, we present a meta-analysis encompassing more than 300 studies and including angiosperms and gymnosperms as well as different biomes (cropland, desert, forest, grassland, tundra, and wetland). The meta-analysis shows that average warming of 2.50 °C (median = 2 °C) significantly increases biomass allocation to roots with a mean increase of 8.1% in R/S. Two factors associate significantly with this response to warming: mean annual precipitation and the type of mycorrhizal fungi associated with plants. Warming-induced allocation to roots is greater in drier habitats when compared to shoots (+15.1% in R/S), while lower in wetter habitats (+4.9% in R/S). This R/S pattern is more frequent in plants associated with arbuscular mycorrhizal fungi, compared to ectomycorrhizal fungi. These results show that precipitation variability and mycorrhizal association can affect terrestrial carbon dynamics by influencing biomass allocation strategies in a warmer world, suggesting that climate change could influence below-ground C sequestration.

Terrestrial carbon (C) dynamics primarily depend on plant carbon economics[1–3]. Plants allocate carbon to their above- and belowground organs, often studied as a ratio between root and shoot biomass (R/S)[4]. In general, the R/S reflects an optimal allocation of resources by plants and is a crucial parameter for estimating terrestrial C storage[5]. When global change factors induce alteration of resource supply for plants, their optimal allocation patterns may shift, resulting in deviations in R/

S[6]. For example, drought and elevated $CO_2$ generally enhance the biomass allocation of roots over shoots for acquiring water and nutrients[7], whereas increased precipitation and nitrogen deposition usually elevate the biomass allocation to shoots for an increased light competition[8,9]. Climate warming has also been shown to affect net primary production in terrestrial ecosystems[10–12], such as by increasing biomass production. If this triggers soil nutrient deficits, plant carbon

[1]Zhejiang Tiantong Forest Ecosystem National Observation and Research Station, School of Ecological and Environmental Sciences, East China Normal University, Shanghai 200062, China. [2]Northeast Asia ecosystem Carbon sink research Center (NACC), Center for Ecological Research, Key Laboratory of Sustainable Forest Ecosystem Management-Ministry of Education, School of Forestry, Northeast Forestry University, Harbin 150040, China. [3]School of Ecology and Environmental Sciences, Yunnan University, Kunming, China. [4]Environmental Futures Research Institute, School of Environment and Science, Griffith University, Nathan, QLD 4111, Australia. [5]Institute of Ecology and Evolution and Oeschger Centre for Climate Change Research, University of Bern, 3012 Bern, Switzerland. ✉e-mail: xhzhou@des.ecnu.edu.cn

economics are likely to change[13–15]. Accordingly, warming effects on R/S may depend on temperature effects on resource supply, which makes it challenging to predict warming effects on plants' biomass allocations, particularly across different environmental conditions[16,17].

The antecedent climatic conditions (e.g., mean annual temperature and precipitation) of a given area are often considered as the significant predictors of R/S variation across terrestrial plants[4]. However, whether the long-term correlation between R/S and climatic conditions conform to immediate R/S response to warmed environments remains unknown[18,19]. Apart from local climate, belowground properties of ecosystems (e.g., nutrients, root traits, and root-soil interface) can further influence the response of R/S to warming[12,20,21]. For instance, the mutualistic symbiosis between root and mycorrhizal fungi (MF), which is universally present in terrestrial ecosystems[22], might affect patterns of plant carbon allocation depending on different nutrient foraging strategies of mycorrhizal root systems[15,23,24]. Plants with the same mycorrhizal association might respond in (ecologically) similar ways to global change factors, especially to alteration of resource availability in soils[25]. As two major types of mycorrhizas, arbuscular mycorrhizal fungi (AMF)'s hyphae penetrate the cell wall and invaginate the cell membrane within the host plant root, whereas the hyphae of ectomycorrhizal fungi (EMF) do not penetrate individual root cell. Moreover, AMF can increase the host-plant's uptake of inorganic nutrients, while EMF provides their host plants with greater access to soil organic nutrient pools[26,27]. Studies have shown that the dominance of a given mycorrhizal fungi type (MFT) could vary across terrestrial biomes, which might explain the variable biomass allocations reported in different biomes[28,29]. Consequently, we suspect that warming effects on R/S across terrestrial biomes could also be regulated by biome-specific MFT dominance.

Here, we propose three potential hypotheses to better understand how biomass allocation in response to warming would alter the distribution of terrestrial carbon at a global scale[30]. First, biomass allocation between above- and belowground compartments of plants may differ due to warming. Using a meta-analysis, we test this hypothesis by examining how the intercept between R/S at warmed temperature and R/S at ambient temperature deviates from 0—we refer to it as a shift in vertical biomass allocation in terrestrial plants (Fig. 1a, b). For instance, when this intercept >0, it would imply that greater biomass is allocated downwards (i.e., vertically), meaning that there will be a greater R/S under warming than that at ambient temperature (Fig. 1a, b). Second, warming can homogenize R/S across biomes. To test this, we examine how the slope between R/S at warmed temperature and R/S at ambient temperature (as a proxy of biomes) deviates from 1—we refer to it as a shift in horizontal variability of R/S in terrestrial plants across biomes (Fig. 1c, d). For instance, when this slope < 1, it would imply that warming decreases the horizontal variability of R/S and thereby causes the homogenization of R/S among diverse biomes (Fig. 1c, d). Third, the mean annual precipitation (MAP) would be a key factor determining the response of R/S due to the direct dependency of belowground resources, such as nutrients and soil water, on MAP. To test these hypotheses, we collected results from 322 warming experiments (Supplementary Fig. 1) with 94 pairs of observations to explore the effects of warming on R/S (Fig. 1f). Using a meta-analytic technique, we show that, with a mean magnitude of 2.50 °C and the median of 2 °C (Supplementary Table 1), climate warming increases R/S (intercept > 0), whereas it decreases the variability in R/S among terrestrial biomes (slope < 1, Fig. 1e). These results suggest that plants with greater investment to shoot in ambient condition shift their allocation strategy by allocating more to root biomass in warmed conditions rather than to shoot biomass, and vice versa. This subsequently triggers warming-induced homogenization of R/S among diverse biomes (Fig. 1e). We further show that MAP and mycorrhizal association with plants correlate strongly with these patterns of R/S in terrestrial plants.

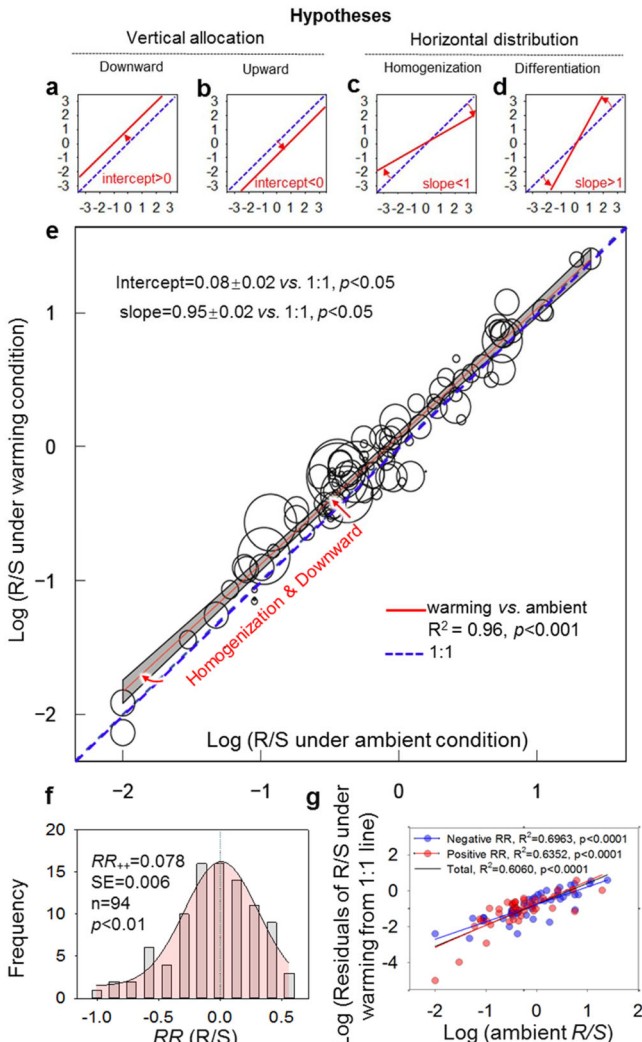

**Fig. 1 | Hypotheses and actual warming effects on root:shoot ratio.**
**a–d** Hypotheses of the warming effect on root:shoot ratio (R/S). **e** The actual relationship between log10-transformed R/S at experimental warming and ambient temperature. **f** The frequency distribution of response ratio of R/S. **g** The log10-transformed residuals of R/S under the warmed condition from 1:1 line along with log10-transformed R/S under ambient condition. The error bands in panel **e** represent the upper and lower 95% confidence intervals. Across studies in our meta-analysis, the intercept in panel **e** was larger than 0, implying greater biomass is allocated (vertically) downwards, i.e., greater R/S under warming than that at ambient temperature (**a**, **e**); the opposite scenario of downward allocation (i.e., upward allocation, intercept smaller than zero) is illustrated in panel (**b**). When the slope is smaller than 1, it would imply that warming decreases the horizontal variability of R/S and thus causes the homogenization of R/S among diverse biomes (**c**, **e**); the opposite scenario of homogenization (differentiation, slope >1) is illustrated in panel (**d**). The size of each dot in panel **e** indicates the relative weight of the individual response ratio of R/S. The $p$-value of the normal distribution test of $RR$(R/S) was 0.3818. The $RR_{++}$ indicates the weighted response ratio of R/S in panel (**f**); the 'n' in panel (**f**) is the sample size of the response ratio of R/S.

## Results

Based on 94 pairs of observations of root:shoot ratio (R/S), average warming of 2.5 °C increased the R/S by 6.1 to 8.8% (with a mean increase of 8.1%), which was significantly larger than 0 ($p < 0.01$, Fig. 1f). The intercept between R/S at warmed temperature and that at ambient temperature ranged from 0.044 to 0.107 (CI > 0, $p < 0.05$, Fig. 1e), suggesting higher biomass allocation to roots in vertical dimension in warmed environments (Fig. 1a). Meanwhile, the slope between R/S at warmed temperature and that at ambient temperature ranged from

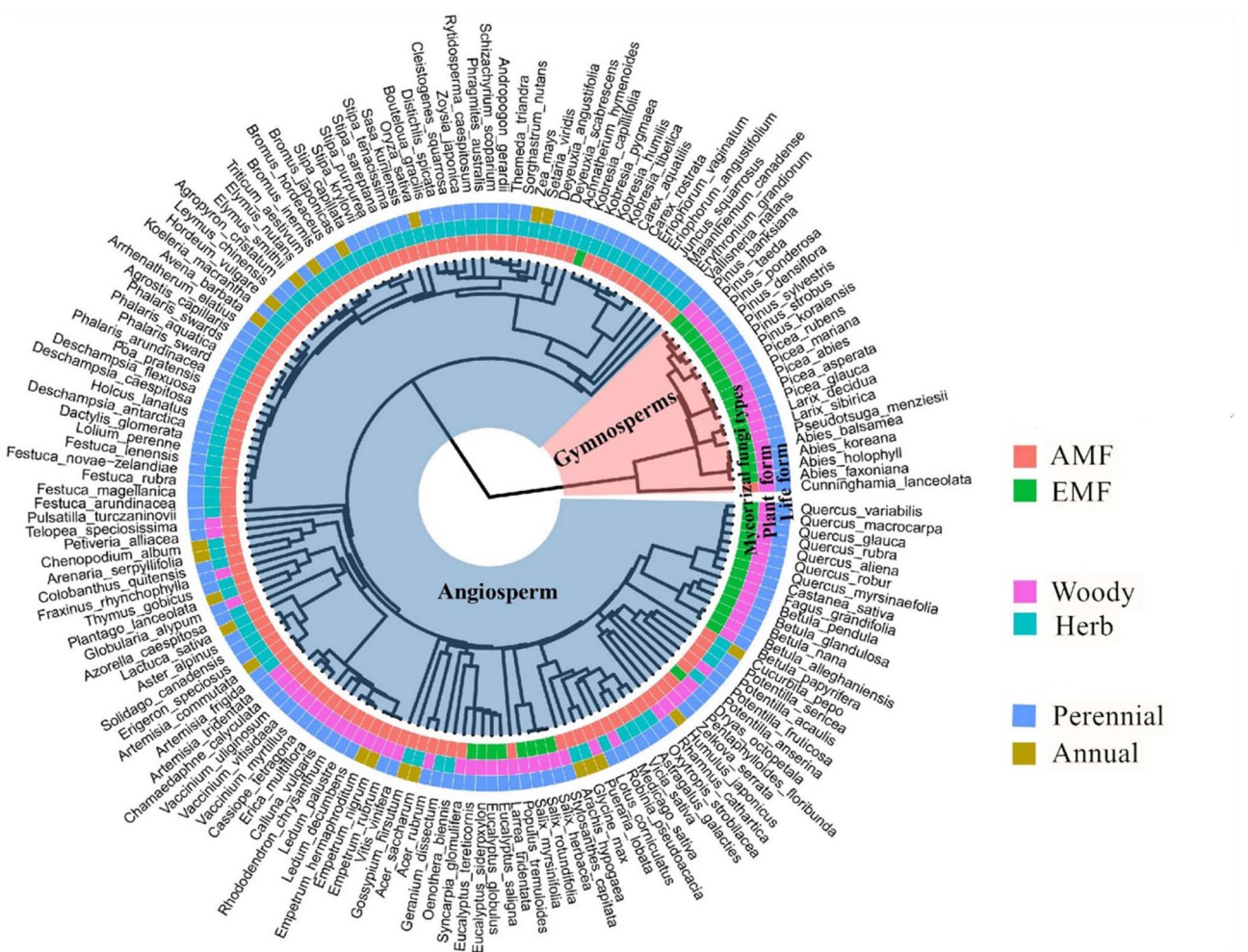

**Fig. 2 | Phylogenetic tree of plant species used in this study.** This phylogenetic tree is based on 164 vascular plant species (consisting of both gymnosperms and angiosperms), which were included to estimate the weighted response ratio of root:shoot ratio, plant total biomass, above- and belowground biomass to warming.

Plants in the phylogenetic tree are shown with their mycorrhizal association (AMF arbuscular mycorrhizal fungi, EMF ectomycorrhizal fungi), life form (woody or herb), and growth form (annual or perennial).

0.908 to 0.997 (CI < 1, $p < 0.05$, Fig. 1e), indicating a horizontal homogenization of R/S among diverse biomes under warmed conditions (Fig. 1e). The biomes with higher R/S in ambient condition had greater variation in R/S in response to warming (Fig. 1g and Supplementary Fig. 2).

On the global scale, 164 species of vascular plants belonging to angiosperm (144) and gymnosperm taxa (20) were included in this study. Relative to annual species (23), perennial ones (141) were more common, whereas the numbers of woody (77) and herb species (87) were comparable in our meta-analysis (Fig. 2). Although the responses of R/S to warming across all the vascular plants displayed a significant phylogenetic signal (Blomberg's K > 1, Supplementary Table 2), mean annual precipitation (MAP) was the most critical factor in determining the warming effects on R/S [i.e., response ratio ($RR$) of R/S, $RR$(R/S)] compared to all other predictors used in our analysis (Fig. 3). We found that MAP was negatively correlated with $RR$(R/S) ($R^2 = 0.104$, Figs. 3 and 4, Supplementary Table 3). Moreover, the weighted $RR$(R/S) reversed from positive to negative in sites with MAP higher than 900 mm (Fig. 4).

Mycorrhizal fungi type (MFT: 121 for AMF and 43 for EMF, Fig. 2) was the second most important factor explaining warming effects on plant's R/S (Fig. 3a). Besides MFT, no significant differences were found in any plant biomass variables (including R/S) in response to warming between evergreen vs. deciduous trees, broad vs. coniferous leaf trees, annual vs. perennial plants, and angiosperms vs. gymnosperms (Supplementary Table 4). Although there was a significant interaction between MFT and biomes (Supplementary Table 5), MFT still explained a greater proportion of variance for $RR$(R/S) relative to biome types (i.e., forest, grassland, cropland, tundra, and wetland, Fig. 3b). The sensitivity of $RR$(R/S) to MAP (i.e., the slopes in Fig. 4b) in biomes with ectomycorrhizal fungi (EMF) was smaller than in biomes with arbuscular mycorrhizal fungi (AMF). Moreover, both warming magnitude and warming duration didn't affect the $RR$(R/S) across warming experiments (Fig. 3a and Supplementary Table 6).

Overall, warming enhanced plant total biomass (TB, +10.0%) mainly by increasing belowground biomass (BGB, +13.1%) even when it decreased aboveground biomass of plants (AGB, −8.9%, $p > 0.05$, Fig. 5). Warming did not change TB or BGB of plants associated with AMF but decreased their AGB by 13.8% and subsequently increased R/S by 9.9%. Warming stimulated TB, AGB, and BGB in biomes with EMF and with mixed AMF and EMF (AM-EMF), but only enhanced R/S in biomes with EMF-associated plants (Fig. 5). MFTs were significantly associated with the warming response of AGB. In contrast, the effects of biomes and plant functional types (PFTs) were true only for the response of BGB (Supplementary Table 5). In our path model, we further found that $MFT_S$ (i.e., AMF, EMF, and AM-EMF) explained 22.9% variation in AGB's response to warming, whereas $PFT_S$ (i.e., herbs

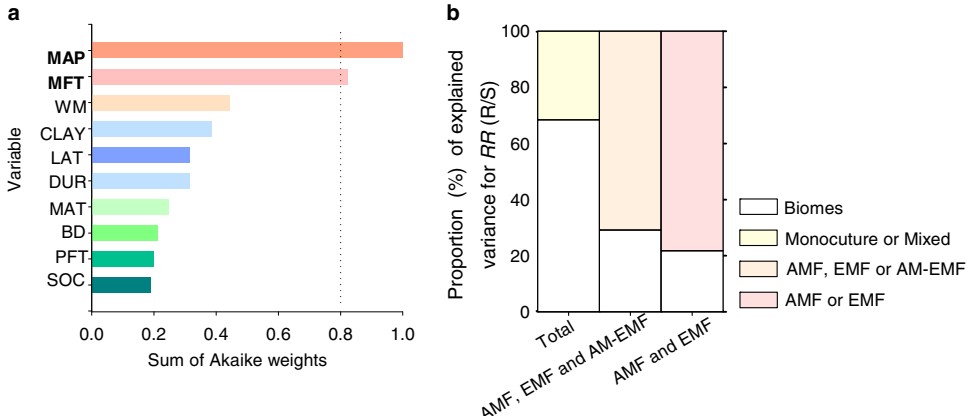

**Fig. 3 | Importance of various predictor variables, and variance explanation of root:shoot ratio in response to warming.** Model-averaged importance of various predictor variables for warming effects on root:shoot ratio [i.e., $RR$(R/S)]. **b** The proportion of total variance of $RR$(R/S) is explained by biomes, plant community composition, and mycorrhizal fungi type (MFT). The variables with an importance value >0.8 were considered as essential predictors. The importance value of a predictor is based on the sum of Akaike weights derived from the model selection using corrected Akaike's information criteria. MFT mycorrhizal fungi type, PFT plant functional type, MAP mean annual precipitation, MAT mean annual temperature, LAT latitude, CLAY the proportion of clay in soils, BD bulk density, SOC soil organic carbon, WM warming magnitude, DUR warming duration (**a**). The CLAY, BD, and SOC were from the topsoil at 0–30 cm soil depth, the detailed information of these predictors is provided in Supplementary Table 3. Biomes include forest, grassland, cropland, tundra and wetland; Monocultures and mixed refer to plant communities with single plant species and multiple plant species, respectively. AMF, EMF, and AM-EMF indicated the biomes with dominant root symbiosis of arbuscular mycorrhizal (AMF), ectomycorrhizal fungi (EMF), mixed AMF, and EMF (AM-EMF), respectively (**b**).

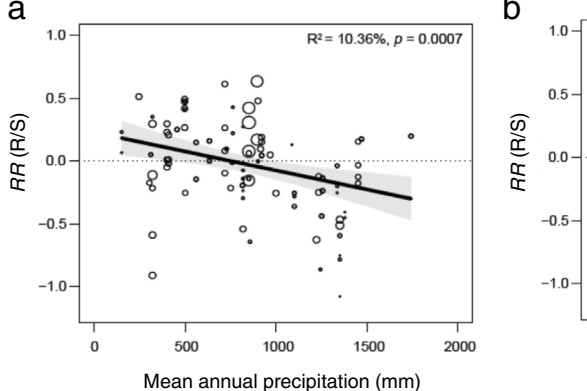
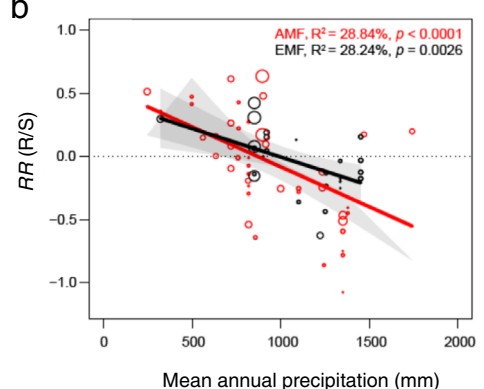

**Fig. 4 | Correlations between mean annual precipitation and response of root:shoot ratio to warming. a** Correlations between mean annual precipitation (MAP, mm) and the response of root:shoot ratio [$RR$(R/S)]. **b** Correlations between MAP and $RR$(R/S) in biomes with dominant root symbiosis of arbuscular mycorrhizal and ectomycorrhizal fungi (AMF and EMF), respectively. The size of the data points indicates relative weights in each plot. The points above the line of $RR$(R/S) = 0 indicate that warming enhanced root biomass accumulation relative to shoot biomass (i.e., $RR$(R/S) > 0), and vice versa. The error bands in panels **a**, **b** represent the upper and lower 95% confidence intervals. The $p$-values were calculated from two-tailed tests.

and woody plants) and MAP were the major significant pathways contributing to 25.4% of variation in BGB's response (Fig. 6). Both responses of AGB and BGB to warming directly contributed (AGB negatively and BGB positively) to shifts in $RR$(R/S) in our path model, explaining a total of 46.3% of variation in $RR$(R/S) across studies (Fig. 6). Moreover, $RR$(R/S) was correlated with BGB responses to warming [$RR$(BGB)] ($R^2$ = 0.31, $p$ < 0.001), but not with AGB responses ($R^2$ = 0.01, $p$ = 0.392, Supplementary Fig. 2). The importance of PFTs on $RR$(R/S) was only true in studies with a warming-induced inhibition of TB (Supplementary Fig. 4). In addition, other experimental factors (e.g., methods of warming and belowground biomass measurement, Supplementary Table 7) and classifications of studies (e.g., field or laboratory experiments, Supplementary Table 4) did not explain any variation in $RR$(R/S).

## Discussion
Our meta-analysis shows that climate warming enhances plants' biomass allocation to belowground but mainly so in drier habitats,

agreeing with the hypothesis of warming-induced vertical downward biomass allocation (Fig. 1)[31]. The contingency of warming effects on plant biomass via soil moisture (Supplementary Fig. 5) might be one of the primary reasons for the upregulation of R/S in many study sites, except for the tundra and alpine biomes where soils are often frozen[17,32]. Even in the studies with decreased total plant biomass in response to warming, we found that belowground biomass allocations were higher than aboveground ones (Supplementary Figs. 6 and 7)[33,34]. Such a plant strategy can have important implications for soil carbon dynamics, potentially due to a lower turnover rate of root-associated carbon (i.e., carbon in hyphae and root per se) compared to shoot-associated carbon[33,34]. Biomes with lower original R/S at ambient conditions showed greater allocation to belowground biomass when warming most likely had induced greater demand for resources[35], agreeing with our hypothesis of horizontal homogenization of R/S across biomes in a warmer world (Fig. 1). We suspect that the upregulated R/S due to warming might alter the carbon transfer between atmosphere and soil[21,36].

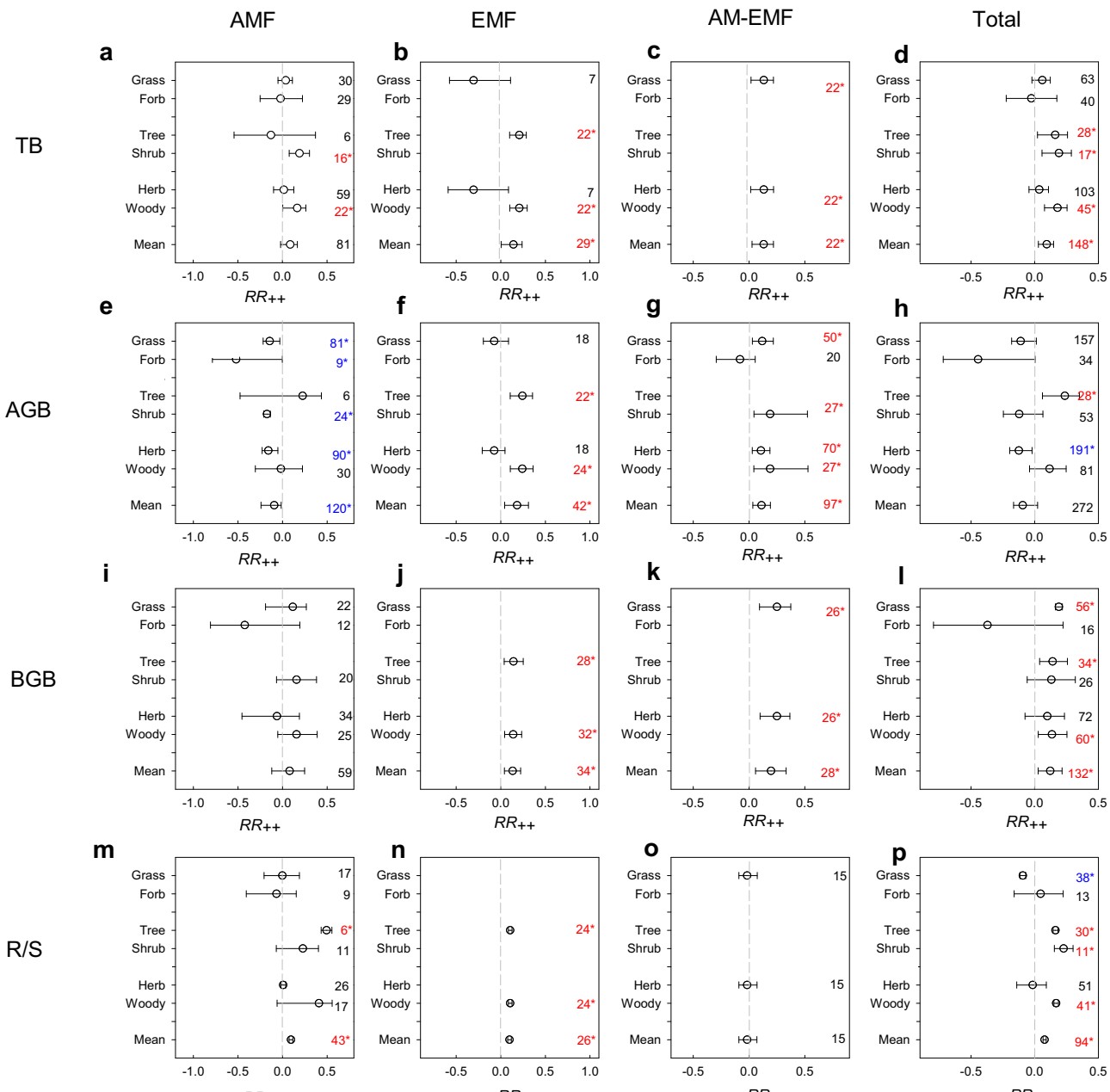

**Fig. 5 | Weighted response ratio of variables to warming in different biomes.**
**a**–**d** Weighted response ratio ($RR_{++}$) of plant total biomass (TB). **e**–**h** $RR_{++}$ of aboveground biomass (AGB). **i**–**l** $RR_{++}$ of belowground biomass (BGB). **m**–**p** $RR_{++}$ of root:shoot ratio (R/S). AMF (**a**, **e**, **i**, **m**), EMF (**b**, **f**, **j**, **n**), and AM-EMF (**c**, **g**, **k**, **o**) indicate biomes with dominant root symbiosis of arbuscular mycorrhizal fungi (AMF), ectomycorrhizal fungi (EMF), mixed AMF and EMF (AM-EMF), respectively (Supplementary Data 1). The numbers inside plots indicate sample sizes. Only the subgroups with a sample size >5 were included. The error bars indicated 95% confidence intervals (CI). When the sample size was larger than 20, the 95% CI was calculated as $RR_{++} \pm 1.96[S(RR_{++})]$, whereas when the sample size was lower than 20, we obtained 95% CI using the bootstrapping method. If 95% CI of effect sizes did not overlap with zero, the warming effect was considered to be statistically significant, and the symbol * with the blue and red color indicated statistical significance of increase and decrease, respectively, of variables in warmed conditions compared to control conditions (*p*-value < 0.05).

More specifically, our results suggest that in areas with mean annual precipitation (MAP) lower than ~900 mm, warmed air and soil surface could induce greater belowground biomass allocation potentially for plants to exploit resources in deeper soil layers, where water is more readily available, and temperature is cooler (Fig. 7a)[37]. The long-term adaptation of plants to water deficiency, e.g., a higher water use efficiency, ensures more biomass accumulation under warmed conditions, thereby triggering an upregulation of R/S (Fig. 7a, and prolonging the C residence time in plants[38]. In areas with MAP greater than ~900 mm, adequate soil moisture could enable positive

effects of warming on nutrient turnover and availability, e.g., soil $NH_4^+$ and $NO_3^-$ for AMF and EMF, respectively (Fig. 7b)[15,39]. Indeed, the allocation to belowground biomass in these areas declined consequently in our meta-analysis, but only for terrestrial plants associated with AMF (Fig. 7b)[40].

The regulation of MAP on R/S under warmed conditions was mainly through the adjustment of belowground biomass accumulation (Supplementary Fig. 3), which responded to variable effects of warming on soil moisture among plant functional types (PFTs, Supplementary Fig. 6)[12,41]. Significant changes in belowground biomass

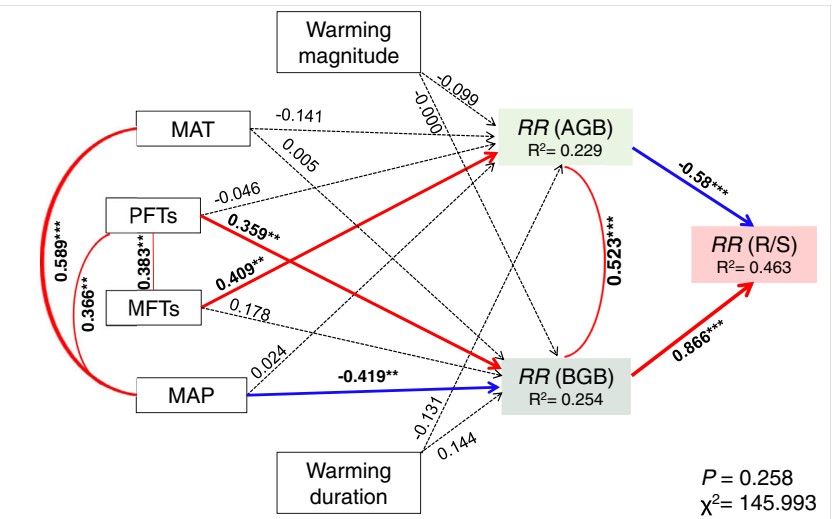

**Fig. 6 | Path analysis examining the effects of plant functional types, dominant mycorrhizal types of root symbiosis, background climate condition, and warming treatment on the response of root:shoot ratio.** PFT$_S$ plant functional types (i.e., herbs or woody plants), MFT$_S$ dominant mycorrhizal fungi types of root symbiosis (i.e., AMF arbuscular mycorrhizal fungi, EMF ectomycorrhizal fungi, or AM-EMF mixed AMF and EMF), MAT mean annual precipitation, MAP mean annual precipitation, *RR*(R/S) response of root:shoot ratio. The solid and dashed lines indicate the significant (*p*-value < 0.05) and non-significant effects (*p*-value > 0.05), respectively, whereas the red and blue solid arrows indicate positive and negative effects (*p*-value < 0.05), respectively. We assigned '0' and '1' for herbs and woody plants, and assigned '0', '0.5', '1' for AMF, AM-EMF, and EMF, respectively, in this analysis (Supplementary Data 2). Symbols *, **, *** indicate statistical significance with *p*-value < 0.05, <0.01, and <0.001, respectively, based on two-tailed tests. In Chi-square test for the path model, the *P*-value was calculated from one-tailed test.

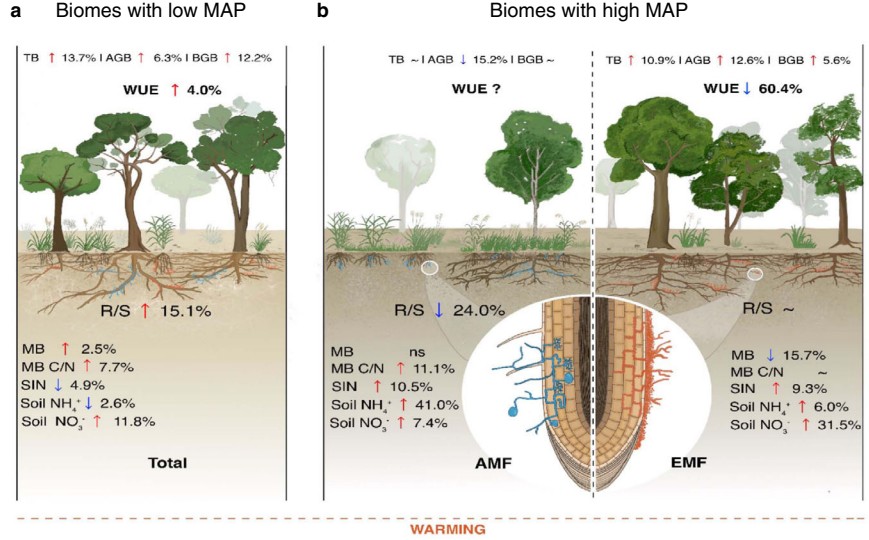

**Fig. 7 | Illustration of our main finding of warming effects on plant total biomass, above- and belowground biomass, and root:shoot ratio. a** Biomes with low mean annual precipitation (MAP) irrespective of mycorrhizal association. **b** Biomes with high MAP and arbuscular mycorrhizal (AMF) or ectomycorrhizal fungi (EMF). TB plant total biomass, AGB aboveground biomass, BGB belowground biomass, MB microbial biomass, MB C/N microbial biomass C N ratio, SIN soil inorganic nitrogen, WUE water use efficiency (Supplementary Data 3, Supplementary Figs. 8 and 9). The up- and downward arrows indicate significant increment and decrement of a response variable (*p*-value < 0.05), whereas "ns" indicates non-significant changes under warmed conditions. "-" corresponds to no change, and "?" indicates not known.

(BGB) in response to warming indeed varied among PFTs compared to other categories used in our meta-analysis (Supplementary Table 5). Woody plants, which are characterized by deeper and evolutionarily mature roots[42], can redistribute soil water by hydraulic lift, supporting an increment in BGB under warmed conditions (Fig. 5)[43]. Herbaceous plants, which are characterized by evolutionarily younger roots relative to woody plants, can potentially adjust root traits (e.g., specific root length) more readily than biomass[12,41]. These warming-induced changes in root traits may have contributed to their non-significant responses in terms of BGB under warmed conditions

(Fig. 5). In addition to MAP, due to interaction between temperature and soil moisture, antecedent mean annual temperature (MAT) was another important background climatic factor influencing the response of R/S in our meta-analysis[44]. For instance, MAT displayed a negative relationship with R/S due to a warming-induced significant increment of aboveground biomass (AGB) in most of the biomes with MAT < 13 °C and plants associated with EMF (Supplementary Table 3)[45].

Plant's association with mycorrhizal fungi further regulated biomass allocation patterns of terrestrial plants in response to warming.

The response of R/S to warming among 164 vascular host plants displayed a significant phylogenetic signal (Fig. 2 and Supplementary Table 2), whereas significant differences were not found between evergreen vs. deciduous trees, broad vs. coniferous trees, annual vs. perennial plants, and angiosperms vs. gymnosperms (Supplementary Table 4). Although both arbuscular mycorrhizal fungi (AMF-) and ectomycorrhizal fungi (EMF-) associated plants exhibited an overall increase in R/S, the underlying mechanisms are likely to be different (Figs. 5, 7). Benefiting from hydrolytic and oxidative extracellular enzymes produced by the hyphae of EMF[46], EMF-associated plants are capable of growing better under nutrient-deficient conditions relative to AMF-associated plants[13,45]. EMF-associated plants showed an increasing trend of AGB, BGB, and R/S under warming (Fig. 5), likely due to hyphal-facilitated soil organic matter (SOM) degradation and root nutrient absorption[47]. By contrast, AMF-associated plants are often more abundant in warmer sites (e.g., in temperate grasslands and deciduous forests) with drier soils[48–51]. Although AMF association correlated with greater water use efficiency in plants under warmed conditions (WUE, Supplementary Fig. 7), AMF-associated plants seemed to slow down the shoot biomass accumulation due to warming-induced soil water deficiency (mainly in grasses, Supplementary Fig. 9)[52] and the potential increment of plant-derived carbon to AMF[53]. Interestingly, the regulation of mycorrhizal fungi types on responses of R/S to warming were mainly related to shoot biomass but not to root biomass (Fig. 6), implying the important roles of mycorrhizal associations on aboveground biomass allocation in terrestrial plants in a warmer world[24,54,55].

As for biomes with mixed AMF and EMF (AM-EMF, Fig. 5, Supplementary Fig. 8), the complementary resource use (e.g., water and nutrients) between plants with arbuscular and ectomycorrhizal fungi provides a plausible explanation for the observed stability of R/S and higher productivity in response to warming (Fig. 4)[56,57]. As we did not find any significant differences in plant biomass allocation (R/S) or biomass itself (AGB and BGB) to warming between evergreen vs. deciduous trees, broad vs. coniferous trees, annual vs. perennial plants, or angiosperms vs. gymnosperms (Supplementary Table 4) when we ignored the types of mycorrhizal fungi in our analysis, it further confirms the importance of plants' association with mycorrhizal fungi in understanding their biomass responses to warming compared to many other plant biotic characteristics.

We suspect that a stronger association of warming effects on biomass allocation with MAP and mycorrhizal associations could potentially lead to a redistribution of R/S in the horizontal dimension with more homogenization and lower variability of biomass allocation patterns among diverse biomes (Fig. 1)[58]. Although we found some clear signals in how warming effects were observed through a greater allocation of root biomass in terrestrial plants relative to shoot biomass, we caution readers about the variability in effect sizes, which could be caused by the distinct effect of warming magnitude and warming duration in a given biome (Supplementary Table 3). For example, warming magnitude might negatively affect the response of R/S for biomes with EMF, while warming duration could induce negative and positive effects for biomes with woody and herbaceous plants, respectively, as revealed in our analysis (Supplementary Table 3).

In summary, climate warming with a mean magnitude of 2.5 °C can enhance belowground biomass allocation as a whole, but mainly so in drier habitats. The warming of 2.5 °C also resembles the predicted IPCC warming (2.4 °C rise of global mean temperature) under a high greenhouse gas emissions scenario (SSP5-8.5) in the following decades (2041–2060)[59], making our results further relevant for the future of terrestrial carbon dynamics. Our results highlight that habitat dryness (or wetness), and types of mycorrhizal association are crucial factors to consider when we aim to predict biomass allocation strategies in terrestrial plants in response to warming across biomes.

## Methods
### Data collection
Using the ISI Web of Science, peer-reviewed journal articles (1950–2020) related to plant biomass, growth, carbon cycling, and/or soil nitrogen under experimental warming were searched with the following specific keywords: (warming OR temperature OR heat* OR greenhouse) AND (biomass OR plant OR allocation OR root OR leaf OR stem OR photosynthe* OR growth OR aboveground OR belowground OR respiration OR soil OR carbon OR nitrogen OR microb*). To avoid bias in publication selection, we used the following criteria to select the studies for our meta-analysis (Supplementary Data 1): (i) Warming treatments were conducted in terrestrial biomes and at least one of the following variables was measured: plant total biomass (TB), aboveground biomass (AGB), belowground biomass (or root biomass, BGB), root:shoot ratio (R/S), water use efficiency (WUE), soil inorganic nitrogen (SIN), soil $NH_4^+$, soil $NO_3^-$, microbial biomass (MB), and/or microbial biomass C/N (MB C/N). (ii) The experimental temperature, warming methods (e.g., open-top chamber, infrared heater, or soil heating cable and greenhouse), and dominant plant species were indicated clearly in both warming and control groups. (iii) Apart from the difference in experimental temperature, other initial environmental conditions and plant species compositions were the same between the control and warming groups. (iv) The duration of the warming experiment was at least longer than one growing season. (v) The mean, standard deviations/errors, and sample sizes of these variables could be extracted from the figures, tables, or the text directly.

Using the above-mentioned five selection criteria, 322 papers associated with the field (280) or laboratory (42) warming experiments were selected (Supplementary Table 4, Notes and References). The methods of air and/or soil warming in selected studies included open-top chamber, infrared heater, soil heating cable, and greenhouse (Supplementary Table 6). The observations (e.g., TB, AGB, BGB, or R/S, WUE, SIN, soil $NH_4^+$, soil $NO_3^-$, MB, and/or MB C/N) in both control and warming groups were extracted using software GetData (version 2.22) as well as treatment and environmental information [e.g., warming duration (DUR) and magnitude (WM), latitude (LAT), mean annual precipitation (MAP) and temperature (MAT)]. If the local climate condition was not reported in a paper, the WorldClim dataset (http://www.worldclim.org/) was used to obtain climate variables based on latitude and longitude for that given study[60]. Roots or belowground biomass measurements across studies included direct harvest, soil core, and ingrowth mesh bags (Supplementary Table 6). In order to analyze the effects of climate (only field studies) and treatment variable (e.g., warming magnitude) on plant biomass responses to warming, we extracted MAP, MAT, DUR, and WM as well as the soil properties (0–30 cm depth), i.e., bulk density (BD), clay content (CLAY), and soil organic carbon (SOC) at each site from Global Gridded Surfaces of Selected Soil Characteristics (BD from IGBP-DIS)[61] and the Harmonized World Soil Database (CLAY and SOC, from version 1.2, https://daac.ornl.gov/SOILS/guides/HWSD.html, Supplementary Table 3). The sites of the selected studies were mainly located in East Asia, North America, and Europe (Supplementary Fig. 1). The MAP and MAT of sites in field experiments ranged from 27 mm to 2400 mm (median 636.2 mm), and −20 °C to 30 °C (median 7.4 °C), respectively. Experimental warming duration and magnitude ranged from one growing season to 25 years (median 2 years), and 0.26 °C to 12 °C (median 2 °C), respectively (Supplementary Fig. 1 and Table 1).

The biome types (including cropland, desert, forest, grassland, tundra, and wetland, Supplementary Table 5), plant functional types (PFTs, including woody plant vs. herb, tree vs. shrub, grass vs. forb, Supplementary Fig. 1), other taxonomy categories (evergreen vs. deciduous tree, broad vs. coniferous leaf tree, annual vs. perennial plant, angiosperm vs. gymnosperm, monoculture vs. mixed plant community, Figs. 2 and 3, Supplementary Table 4) and mycorrhizal

fungi types (MFTs, Supplementary Fig. 1 and Supplementary Table 5) for dominant plants in each case study were confirmed based on the original publications and the latest FungalRoot database (https://nt.ars-grin.gov/fungaldatabases/)[45]. The MFTs of biomes were labeled as AMF, EMF, or AM-EMF if the root symbiosis of dominant plants were arbuscular mycorrhizal fungi (AMF), ectomycorrhizal fungi (EMF), or mixed AMF and EMF (AM-EMF). Most analyses in this study focused on AMF, EMF, and AM-EMF, as biomes with non-mycorrhizal fungi (NMF) were too few to conduct any meta-analysis (Supplementary Fig. 1).

### Data analysis

In this study, we employed response ratio [$RR$, natural log (ln) of the ratio of the mean value in warming treatment ($\bar{X}_t$) to that in control ($\bar{X}_c$), Eq. (1)] to reflect warming effects on various response variables[62,63].

$$RR = \ln\left(\frac{\overline{X_t}}{\overline{X_c}}\right) = \ln(\overline{X_t}) - \ln(\overline{X_c}) \tag{1}$$

The weighted response ratio ($RR_{++}$, Eq. (2)), and the standard error of $RR_{++}$ [s($RR_{++}$), Eq. (3)] in each subgroup [e.g., different MFTs (AMF, EMF, or AM-EMF) or PFTs (woody plants or herbs)] were calculated using individual $RR(RR_{ij})$ and its weight ($w_{ij}$), which is the reciprocal of the variance ($v_{ij}$, Eq. (4), Fig. 4).

$$RR_{++} = \frac{\sum_{i=1}^{m}\sum_{j=1}^{k} w_{ij}RR_{ij}}{\sum_{i=1}^{m}\sum_{j=1}^{k} W_{ij}} \tag{2}$$

$$s(RR_{++}) = \sqrt{\frac{1}{\sum_{i=1}^{m}\sum_{j=1}^{k} w_{ij}}} \tag{3}$$

where $m$ is the number of subgroups, and $k$ is the number of $RR$ in the $i$th subgroup ($i = 1, 2, …, m; j = 1, 2, …, k$).

$$w_{ij} = \frac{1}{\mathbf{V}_{ij}}, \quad v = \frac{S_t^2}{n_t \bar{X}_t^2} + \frac{S_c^2}{n_c \bar{X}_c^2} \tag{4}$$

The $n_t$, $n_c$, and $S_t$, $S_c$ are the numbers of replicates and standard deviations for a given variable in the warming and control group, respectively. The warming effect on a given response variable was considered statistically significant when the 95% CI of the effect size did not overlap with zero. The percentage change of a response variable under warmed conditions was estimated by [exp($RR_{++}$) − 1] × 100%.

The between-group heterogeneity ($Q_b$) of effect sizes was estimated using the Q-statistic in MetaWin 2.1 (Supplementary Table 5)[64]. Stepwise linear regression was used to analyse the impacts of environmental conditions on the responses of root:shoot ratio [$RR$(R/S), Supplementary Table 4]. We used Blomberg's K values to test the phylogenetic signals on the warming response of plant biomass variables using the "picante" package (version 1.8.2)[65] in R (version 4.1.3, R Core Team, 2018, Supplementary Table 2). The effects of biomes, MFTs, PFTs, experimental methods (warming methods and methods to measure root or belowground biomass), warming magnitude (WM), and warming duration (DUR) were examined to explain the variation in response variables using the analysis of variance (ANOVA, Supplementary Tables 5 and 6). Both WM and DUR were sub-grouped into six classes according to the method explained in Lin et al. (Supplementary Table 6)[12]. The importance of each predictor for the response ratio of a given response variable was expressed as the sum of Akaike weights derived from the model selection using the "glmulti" package (version 1.0.8)[66] in R (Fig. 3a, Supplementary Fig. 3). Nested analysis was performed using the "nlme" package (version 3.1-158)[67] in R to estimate the variance explained by MFTs (Fig. 3b). We used the

"metafor" package (version 3.4-0)[68] in R for the linear regression analysis between MAP and $RR$(R/S) (Fig. 4). Structural equation model (SEM) was performed using the "lavaan" package (version 0.6-12)[69] in R to examine the effects of PFTs, MFTs, background climate (MAT, MAP), and warming treatment (WM and DUR) on $RR$(R/S) via changes in AGB and BGB under warming (Fig. 6). The figures to depict correlations between specific response variables were drawn in SigmaPlot 12.5 (Systat Software, San Jose, CA, Supplementary Figs. 2 and 3).

### Reporting summary

Further information on research design is available in the Nature Research Reporting Summary linked to this article.

## Data availability

The data used in this study are available in Supplementary Data 1–3.xlsx.

## Code availability

The code used in this study is available at https://figshare.com/articles/online_resource/Code1_2_txt/20390592/1.

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

## Acknowledgements

We thank Pryianka Ambhavane (https://ambypriya.myportfolio.com/) for her help in drawing Fig. 7. X.Z. is supported by the National Natural Science Foundation of China (Grant No. 31930072). L.Z. is supported by the National Natural Science Foundation of China (Grant No. 32071593, 31600352), the National key Research and Development Program of China (Grant No. 2020YFA0608403), and the Fundamental Research Funds for the Central Universities. M.P.T. is supported by the Swiss State Secretariat for Education, Research, and lnnovation (SERI) under contract number M822.00029.

## Author contributions

L.Z. collected and analyzed the data and wrote the manuscript with substantial contributions from M.P.T. and X.Z. X.Z. conceived, designed, and oversaw the study. L.Z. and Y.H. did the statistical analysis with suggestions from M.P.T. Y.H., Y.F., Z.D., M.L., X.S., C.L., C.L., R.L., G.Z., and S.H.B. commented on the manuscript.

## Competing interests

The authors declare no competing interests.
