## [Peer Review File · Nature Communications]

Reviewers' Comments:

Reviewer #1:

Remarks to the Author:

Overall I think this paper is a great contribution with some problems that need to be addressed before it can really shine.

In summary: some sentences were difficult for me to parse, some citations feel inappropriate (cited study not general enough to support the authors' statements). More transparency is needed in regards to the rationale for the author's hypotheses and details of their meta-analysis. I also strongly advise that the figures are changed so that the residual (differences from the 1:1 line) log response ratios are plotted. At present, large effect sizes are too difficult to visualize in the plots.

I feel all these comments, which I spell out in more detail below, are amenable to revision. Moreover, this paper is an original and broad contribution in the study of how plants with different mycorrhizal associations will respond to global change.

Detailed comments:

Abstract:

Can you clarify what the % effect size refers to here (e.g., % for every 1 degree increase in temperature). "Warming" is vague. As you have a response ratio here, you may need to define the median / mean range of warming that resulted in this response ratio.

Introduction

L37-50

Opening paragraph is a bit of a mish-mash – doesn't give a broad-scope one would expect at the beginning of a paper addressing a fundamental question about plant carbon economics.

L46:50.

It is not clear how the increase in NPP should inform the R:S ratios mentioned earlier – if resource demand increases but the marginal benefits of above and below-ground benefits follow the same ratio, then you would expect no differences in R:S ratios with changes in the magnitude of NPP. If on the other hand the increase in NPP is caused by a greater deficit in nutrients (than in light), then you would expect higher R:S. As it is, this sentence does not address this.

L51:54

This sentence is too difficult to parse. Make it easier on your reader.

L62-24: "Plants with the same mycorrhizal association generally act in ecologically similar ways in the face of changing environments, especially to alteration of resource availability in soils²¹"

This is a very general statement. However, Ref 21 is

Thomas, H. J. D. et al. Traditional plant functional groups explain variation in economic but not size-related traits across the tundra biome. *Global Ecol. Biogeogr.* 28, 78-95, doi:10.1111/geb.12783 (2018).

The cited paper is insufficient to support the authors' statement. For starters, it is focusing on trait distributions in one biome. "In the face of changing environments..." makes it vague whether we are talking about temporal climate changes or spatial climatic-trait associations among functional groups.

2 more general papers to check out:

Averill et al., 2019. *Global Change Biology*.

Steidinger et al., 2019. *Nature*.

Methods

I keep getting hung up on what "warming" means. I understand the format of these papers might mean that methodological details are mostly listed after the Intro/Results/Discussion text. However, it is making it hard for me to understand what is being compared.

Some additional details for how the meta-analyses are carried out would help (even if one or two sentences is all the word-mass you can allocate to this). Could you include some summary data (# of studies, median/mean magnitude of warming, etc.).

Figures

The authors present the effect size as the difference with the 1:1 line (warming vs. no warming). Thus, the x and y axis are on the same scale (spanning the inter-experimental variability in log response ratios). Because the intra-experimental variability to warming is smaller in magnitude, this forces the reader to discriminate what sound like large effect sizes (14%) as minuscule differences from the 1:1 line.

This is easily remedied. Simply plot the residuals from the 1:1 line, allowing the y-axis to fall along a different scale than the x.

Also, I do appreciate that Figure 1 gives a graphical overview of your hypotheses. However, it gets more complicated when there are combination of intercept and slope outcomes. For example, if you fit a positive intercept (downward vertical allocation) and a slope < 1 (homogenization), then what you have is different responses based on the ambient R/S ratio. You could have plants that invest primarily in shoots under ambient conditions (negative log ratio) showing greater increase in roots than plants that invest primarily in roots (positive log ratio). Any description of the overall response would have to include both terms.

Figure 2: can we annotate what $RR > 0$ and $RR < 0$ mean in simple interpretation terms.

L94-97:

"On average, warming significantly increased root: shoot ratio (R/S) by 6.7% with the confidence interval (CI) of intercept of 0.044-0.107 (Fig. 1e), which was significantly larger than 0 ($p < 0.05$), agreeing with the assumption of more biomass allocation to roots in vertical dimension in warmed environments."

This sentence is a mess. First, there is the repetition – "significantly increase", "CI was significantly larger than 0 ($p < 0.05$).". Then this is stated as "agreeing with the assumption." However, your assumption is actually a hypothesis. Clean it up.

Reviewer #2:

Remarks to the Author:

The topic of this manuscript is very relevant to Nature Communications in that the study reviews the global literature to assess how warming under climate change may impacts root-to-shoot ration of different types of plant. It is recommended that the manuscript is suitable for publication after some major revisions as outline below:

Major comments:

Methodology:

From the Abstract also the brief description of methodology (e.g. L88), it is not clear:

(a) what sort of studies were collated in the review. It is not clear from reading the main report what sorts of experiments were collated. Were these glass house studies, or field studies, or both?
(b) What sort of plants were included in the review. This also is not clear when reading the first part of the 'Data collection' section of the extended Methods section. What plant functional types were reviewed is not made clear until the plant functional types categories assess were introduced in L32-33 of 'Data collection' section of the extended Methods.

Objectives:

L81-2. This objective has not been as well introduced in the previous paragraphs. Why would we

expect warming to homogenize R: across the biomes? The link between this 2nd hypothesis and the arbuscular mycorrhizal fungi and ectomycorrhizal fungi is not clearly made. L116-117. First time this part of the methodology is introduced. Suggest making assessing the attribution for change in R:S with warming a 3rd objective. For example, how were the results presented in Fig. 2 obtained? There is no description of this included.

Results and Discussion:

L230-237. These statements here in the Conclusions, and in the Discussion, are too strong given the variability in the collated data which is clear in Fig. 1 and 3. For example, in Fig. 3, the differences between AMF and EMF do not seem significantly different, although each could have a significant fitted relationship fitted.

L94. Unclear how the significant increase in R:S of 6.7% is calculated from Fig. 1e.

Structure, or layout of the article:

Suggest refining objectives as discussed above. Then arrange the article to report on these specific objectives. For example,

- L170. The discussion of SOC implications in the first paragraph of the Discussion seems misplaced. Why not start with the findings in relation to your key objective first, then discuss other possible feedback effects after that?

- L180. Ditto for soil nutrient effects. This seems very speculative, and not directly related to the key objectives anyway.

Minor comments

Fig. 2. Need to state what soil layers (or depth) is being considered in the SOC and BD variables, e.g. 0-30 cm, or 0-50 cm etc? Also, were the MAP and MAT data collated from the same global data source? How accurate were these estimates?

Fig. 3. What are the grey shaded areas? Please include description in the figure caption.

Reviewer #3:

Remarks to the Author:

This manuscript describes results from a meta-analysis of over 300 studies showing that warming significantly increases biomass allocation to roots (13%), and that mean annual precipitation of the site, and the type of mycorrhizal fungi were mostly related to this response. In particular, this response was greater in drier than wetter sites and was more significant for plants associated with arbuscular mycorrhizal (AM) fungi than those with ectomycorrhizal (EM) fungi. These results are interesting, but not of special novelty.

The first major finding was that "climate warming enhances belowground biomass allocation in plants but mainly so in relatively drier habitats". This finding is solid, but is largely expected. Warming often increases evapotranspiration and water stress on plants, and plants respond to this change by allocating more photosynthates belowground to exploit more water, particularly in drier habitats.

The second major finding was that there was a difference in plant biomass allocation responses between plants with arbuscular mycorrhizal fungi and those with ectomycorrhizal fungi. This seems to be a little more interesting, but the discussion of the potential underlying mechanisms was largely speculative and highly questionable. The manuscript attempts to attribute this difference to different nutrient acquisition strategies between the two types of mycorrhizae, but ignores the fundamental differences in their host plants, particularly in terms of root systems and leaf structure. Host plants for ecto-mycorrhizal (ETM-) fungi are largely perennials and dominantly coniferous and woody plants, which are with deep root systems and often with high water use efficiency (e.g., water conserving needles with a thick, waxy coating). These plants are likely less sensitive to changes in water availability induced by warming. In contrast, many plant species associated with arbuscular mycorrhizal fungi are annuals and deciduous, including the vast majority of herbaceous grasses and forbs with shallow root systems. The shallow root system makes many AMF-associated host plants more sensitive to warming-induced moisture changes in

surface soils. In this sense, the coarse comparison between these two types of plants becomes less meaningful because it does not provide sufficient information for readers to infer whether the difference observed stems from different mycorrhizae-types or different hosts.

Response letter to comments (NCOMMS-21-40930-T)

Here are our detailed responses to reviews. Please note that the comments from the reviewers are in *italics* followed by our responses in **regular** text.

Responses to Associate Editor

Thank you again for submitting your manuscript "Warming effects on biomass allocation are jointly regulated by precipitation and mycorrhizal association in terrestrial plants" to Nature Communications. We have now received reports from 3 reviewers and, after careful consideration, we have decided to invite a major revision of the manuscript. As you will see from the reports copied below, the reviewers raise important concerns. We find that these concerns limit the strength of the study, and therefore we ask you to address them with additional work. Without substantial revisions, we will be unlikely to send the paper back to review. In particular, be more transparent in explaining and showing the results of the meta-analysis, also consider including taxonomy as a factor based on Reviewer 3 concerns. You may extend the Methods section as required since this section is not included in the word count. If you feel that you are able to comprehensively address the reviewers' concerns, please provide a point-by-point response to these comments along with your revision. Please show all changes in the manuscript text file with track changes or colour highlighting. If you are unable to address specific reviewer requests or find any points invalid, please explain why in the point-by-point response.

[Response] We are very thankful for your positive assessment of our manuscript. We further appreciate the chance to revise it based on the very constructive suggestions from the three reviewers. We have incorporated several suggestions, including your suggestion to consider new analyses based on the comments from the reviewer 3 regarding the effects of plant taxonomy as well as other variables (e.g., biomes and plant functional groups) to explain the variation in effect sizes. All these new analyses have made our results clearer, and we followed your suggestions to extend the method section as a consequence of additional analyses. Finally, while the submitted version is a substantial revision, the main conclusion from the original version has not changed, which is: warming effects on plant biomass allocation above- and belowground depend on site wetness and mycorrhizal association of a plant.

Responses to Reviewer #1

Overall I think this paper is a great contribution with some problems that need to be addressed before it can really shine. In summary: some sentences were difficult for me to parse, some citations feel inappropriate (cited study not general enough to support the authors' statements). More transparency is needed in regards to the rationale for the author's hypotheses and details of their meta-analysis. I also strongly advise that the figures are changed so that the residual (differences from the 1:1 line) log response ratios are plotted. At present, large effect sizes are too difficult to visualize in the plots. I feel all these comments, which I spell out in more detail below, are amenable to revision. Moreover, this paper is an original and broad contribution in the study of how plants with different mycorrhizal associations will respond to global change.

[Response] We are grateful for the positive remarks by the reviewer. Agreeing with these constructive and helpful suggestions, we have thoroughly revised our manuscript, which has substantially improved the manuscript. For example, we rephrased several sentences to increase the clarity (e.g., L22-28, 40-47, 69-72, and L230-244). We also made sure that all the cited references align with our statements. For example, “Averill, C., Dietze, M. C. & Bhatnagar, J. M. Continental-scale nitrogen pollution is shifting forest mycorrhizal associations and soil carbon stocks. *Global Change Biol.* 24, 4544-4553, doi:10.1111/gcb.14368 (2018)” and “Steidinger, B. S. et al. Climatic controls of decomposition drive the global biogeography of forest-tree symbioses. *Nature* 569, 404-408, doi:10.1038/s41586-019-1128-0 (2019)” were used to replace the previous reference “Thomas, H. J. D. et al. Traditional plant functional groups explain variation in economic but not size-related traits across the tundra biome. *Global Ecol. Biogeogr.* 28, 78-95, doi:10.1111/geb.12783 (2018)”.

We revisited our main hypotheses and clarified them further to streamline the manuscript well (L87-99). We further provide a more detailed information on plants and experimental methods to make the Method section more transparent for the data analysis (e.g., L11-16, 22-25, 32-44, 45-51, 87-89 in Method). Finally, in line with the reviewer's suggestion, we provide a more detailed outline of the warming effect sizes on plant biomass allocation by showing it across functional groups and life forms of plants (Figure 5 in the revised manuscript, pasted below as Figure RF1). In addition, the frequency distribution of responses ratio of root: shoot ratio (RR (R/S)) were added to the revised Figure 1f (below as Figure RF2f), showing the weighted response ratio of R/S to better visualize the range of the effect sizes. We also added a new plot to show residuals of R/S under warming condition from 1:1 line along with R/S under ambient condition to previous Fig. 1 (blow as Figure RF3 g, Fig. 1 g in the main manuscript), which suggested that plants with greater R/S in ambient condition had greater variation in R/S in response to warming (results added to the revised version, L124-125).

Figure RF1 (Figure 5 in the revised manuscript): Weighted response ratio (RR_{++}) of total biomass (TB, a-d), above- and belowground biomass (AGB and BGB, e-h and i-l) and root: shoot (R/S, m-p) in biomes with dominant root symbiosis of arbuscular mycorrhizal (AMF, a, e, i, and m), ecto-mycorrhizal fungi (EMF, b, f, j, and n), and biomes with mixed root symbiosis of mycorrhizal fungi (Mixed, c, g, k and o), and in total studies (Total, d, h, l, and p).

Figure RF2 (Figure 1 in the revised manuscript): Hypotheses of warming effects on root: shoot ratio (R/S, a, b, c, d) and the actual relationship between log₁₀-transformed R/S with experimental warming and that at ambient temperature (e), and the frequency distribution of response ratio of R/S (RR(R/S), f), and the frequency distribution of response ratio of R/S (RR(R/S), f), and the residuals of R/S under warming condition from 1:1 line along with R/S under ambient condition (g).

Abstract: Can you clarify what the % effect size refers to here (e.g., % for every 1 degree increase in temperature). “Warming” is vague. As you have a response ratio here, you may need to define the median / mean range of warming that resulted in this response ratio.

[Response] The reviewer makes a good suggestion. The range of warming magnitude (WM) across all the studies in our meta-analysis varied from as small as 0.26 °C to as large as 12 °C (Fig. S1e in the revised manuscript, Table S1 in the revised manuscript; pasted below as Figure RF3, RT1) with the mean of 2.5 °C, and the median of 2 °C. We added this information to various parts of the manuscript (abstract (L26-29), Introduction (L91-92), and the Results (L117-119) sections).

Figure RF3 (Figure S1 in the revised manuscript): Site location (a) and frequency of warming studies (b-e) included in this meta-analysis. Frequency of the studies was analyzed based on the mean annual precipitation (MAP, mm, b) and temperature (MAT, °C, c) in sites as well as warming duration (y, d) and magnitude (°C, e) for different plant functional types (i.e., tree, shrub, grass and forb). MFT--PFT indicated biomes with mycorrhizal fungi types and plant functional types; AMF, EMF, NMF, and AM-EMF indicated biomes with dominant root symbiosis of arbuscular mycorrhizal, ecto-mycorrhizal, non-mycorrhizal, and mixed arbuscular and ecto-mycorrhizal fungi. The numbers in the parentheses were the study frequency in each MFT in plot a.

Table RT1 (Table S1 in the revised manuscript): Ranges of predictor variables for warming responses of biomass allocation.

Predictor variables	Unit	Range	Median
Mean annual precipitation (MAP)	mm	27 ~ 2400	636.2
Mean annual temperature (MAT)	°C	-20 ~30	7.4
Warming magnitude (WM)	°C	0.26 ~12	2.0
Experimental duration (DUR)	GS or year	One GS ~ 25	2.0
Latitude	°	-68.02~78.9	41.3
Soil organic carbon (SOC)	g/kg	0.17 ~ 33.63	1.2
Bulk density (BD)	g/cm ³	0.76 ~ 1.66	1.4
Clay	%	4 ~ 62	19.0

Notes: MAP, with spatial resolution of 0.5°; MAT, with spatial resolution of 0.5°; GS, growing season; SOC, 0-30 cm depth from version 1.2 HWSD, with spatial resolution of 6%-46% 30 arc-second; BD, 0-30 cm depth from IGBP-DIS, with spatial resolution of 0.08333°; Clay, soil clay content, 0-30 cm depth, from version 1.2 HWSD, with spatial resolution of 6%-46% 30 arc-second.

Introduction L37-50. Opening paragraph is a bit of a mish-mash – doesn't give a broad-scope one would expect at the beginning of a paper addressing a fundamental question about plant carbon economics.

[Response] We agree with the reviewer, and have revised the start of the introduction as (L 38-43): “Terrestrial carbon (C) dynamics primarily depends on plant carbon economics¹⁻³. Plants allocate carbon to their above- and belowground organs, which is often studied as a ratio between root and shoot plant biomass (aka R/S)⁴. In general, the R/S reflects an optimal allocation of resources by plants, and is a key parameter for estimating terrestrial C storage⁵. When global change induces alteration of resource supply for plants, their optimal allocation patterns may shift, resulting into deviations in R/S⁶”.

In addition, we concluded the first paragraph to highlight a major research gap, which we aim to address in our manuscript (L 47-52): “Climate warming has also been shown to affect net primary production in terrestrial ecosystem, such as by increasing biomass production⁷⁻⁹, and if this triggers soil nutrient deficits, plant carbon economics are likely to change^{10,11}. Accordingly, warming effects on R/S may depend on temperature effects on resource supply, which makes it challenging to predict warming effects on plant's biomass allocations, particularly at a global scale^{12,13}”.

L46:50. It is not clear how the increase in NPP should inform the R:S ratios mentioned earlier – if resource demand increases but the marginal benefits of above and below-ground benefits follow the same ratio, then you would expect no differences in R:S ratios with changes in the magnitude of NPP. If on the other hand the increase in NPP is caused by a greater deficit in nutrients (than in light), then you would expect higher R:S. As it is, this sentence does not address this.

[Response] The reviewer makes a very good point and we accordingly have clarified it in the revised version as (L 47-49): “Climate warming has also been shown to affect net primary production in terrestrial ecosystem, such as by increasing biomass production⁷⁻⁹, and if this triggers soil nutrient deficits, plant carbon economics are likely to change^{10,11}”.

L51:54, *This sentence is too difficult to parse. Make it easier on your reader.*

[Response] Done as suggested (L53-55): “The antecedent climatic conditions (e.g., mean annual temperature and/or precipitation) of a given area are often considered as the major predictors of R/S variations across terrestrial plants⁴”.

L62-24: *“Plants with the same mycorrhizal association generally act in ecologically similar ways in the face of changing environments, especially to alteration of resource availability in soils²¹”. This is a very general statement. However, Ref 21 is Thomas, H. J. D. et al. Traditional plant functional groups explain variation in economic but not size-related traits across the tundra biome. Global Ecol. Biogeogr. 28, 78-95, doi:10.1111/geb.12783 (2018). The cited paper is insufficient to support the authors’ statement. For starters, it is focusing on trait distributions in one biome. “In the face of changing environments...” makes it vague whether we are talking about temporal climate changes or spatial climatic-trait associations among functional groups. 2 more general papers to check out: Averill et al., 2019. Global Change Biology. Steidinger et al., 2019. Nature.*

[Response] We thank the reviewer for pointing about the inaccuracy of our citations. Indeed, the two suggested papers about the different traits of plants with ECM and AM better support our statement. We made these changes in the revised version (L62-64): “Plants with the same mycorrhizal association generally respond in (ecologically) similar ways to global change factors, especially to alteration of resource availability in soils^{14,15}”.

Plants with different types of fungal symbiosis are characterized with diverse nutrient economic strategies. For example, ECM plants are more nutrient (N or P) use-conservative relative to AM plants, especially in temperate latitudes (Averill et al., 2019, PNAS). Therefore, we assumed that the plants with the same mycorrhizal association will respond in similar ways to global change factors. We replaced the “changing environments” with “global change factors” here to be more specific about anthropogenic effects on variation in climatic conditions (L63).

Methods. I keep getting hung up on what “warming” means. I understand the format of these papers might mean that methodological details are mostly listed after the Intro/Results/Discussion text. However, it is making it hard for me to understand what is being compared. Some additional details for how the meta-analyses are carried out would help (even if one or two sentences is all the word-mass you can allocate to this). Could you include some summary data (# of studies, median/mean magnitude of warming, etc.).

[Response] We agree with the reviewer that adding details about the warming would ease the readership. The averaged warming magnitude (WM) across all the studies compiled in our meta-analysis was 2.50 ± 1.62 °C (mean \pm sd) with a median of 2 °C (Table S1 and Fig. S1e in Supporting information). Based on the 94 pairs of R/S observations, we found that warming increased R/S by 6.1% to 8.8%. We added this information to the revised abstract (L25-29):

“Using a global synthesis of >300 studies, we here show that mean experimental warming of 2.50 °C (median =2 °C) significantly increase biomass allocation to roots with an increase of 8.1 % (95% confidence interval of 6.1%-8.8%) in R/S.....”, and the last paragraph in the Introduction section (L91-92): “Our results show that, with a mean magnitude of 2.50 °C and the median of 2 °C (Table S1), global warming increases R/S (intercept >0), whereas decreases the variability in R/S among terrestrial biomes (slope <1, Fig. 1e)”. We finally also added to the Results section (L117-119): “Based on 94 pairs of observations of R/S, an average warming of 2.5 °C increased root: shoot ratio (R/S) by 6.1% - 8.8% (with a mean increase of 8.1%), which was significantly larger than 0 (p<0.01, Fig. 1f)”. This averaged 2.5 °C warming value also resembles the predicted IPCC warming (2.4 °C rise of global mean temperature) under high greenhouse gas emissions scenario (SSP5-8.5) in the next decades (2041-2060)¹⁶, making our results further relevant for terrestrial carbon dynamics in a changing world (In L 316- 320).

Figures. The authors present the effect size as the difference with the 1:1 line (warming vs. no warming). Thus, the x and y axis are on the same scale (spanning the inter-experimental variability in log response ratios). Because the intra-experimental variability to warming is smaller in magnitude, this forces the reader to discriminate what sound like large effect sizes (14%) as minuscule differences from the 1:1 line. This is easily remedied. Simply plot the residuals from the 1:1 line, allowing the y-axis to fall along a different scale than the x. Also, I do appreciate that Figure 1 gives a graphical overview of your hypotheses. However, it gets more complicated when there are combination of intercept and slope outcomes. For example, if you fit a positive intercept (downward vertical allocation) and a slope < 1 (homogenization), then what you have is different responses based on the ambient R/S ratio. You could have plants that invest primarily in shoots under ambient conditions (negative log ratio) showing greater increase in roots than plants that invest primarily in roots (positive log ratio). Any description of the overall response would have to include both terms.

[Response] This is an excellent suggestion made by the reviewer. In the revised Figure 1, we added a new plot as *Figure 1 panel g* (shown below as Figure RF4) to show the residuals of R/S in warming condition from the 1:1 line along with those under ambient condition. The deviation of R/S under warming from ambient condition also indicated significant relationship with R/S under ambient condition (Figure RF5), which was as added as Figure S2 in supplementary information. The correlated results were added in L124-125: The biomes with greater R/S in ambient condition had greater variation in R/S in response to warming (Fig. 1g and S2).

In addition, the frequency of individual response ratio of R/S was supplemented with weighted mean response ratio of R/S in the revised Figure 1e (Figure RF2 in this file). The description about the linear regression was revised to highlight the different response trends along with ambient R/S, detailed as (L94-97): “These results suggest that plants with greater investment to shoot in ambient condition shift their allocation strategy by allocating more to root biomass in warmer conditions rather than to shoot biomass, and vice versa, which subsequently triggered warming-induced homogenization of

R/S among diverse biomes (Fig. 1 e)”.

Figure RF4 (Figure 1g in the revised manuscript): The residuals of R/S under warming condition from 1:1 line along with R/S under the ambient temperature.

Figure RF5 (Figure S2 in the revised manuscript): The deviation (a) and absolute deviation (b) of root/shoot ratio (R/S) under warming from ambient condition along with R/S under ambient condition. Negative and Positive RR indicates the points with negative and positive response ratio of R/S, respectively.

Figure 3: can we annotate what $RR > 0$ and $RR < 0$ mean in simple interpretation terms.

[Response] The dotted lines in Figure 3 a and b (the previous version) indicated the $RR (R/S) = 0$. The points above the line ($RR = 0$) meant $RR (R/S) > 0$ indicated that warmer condition enhanced the allocation to root biomass relative to shoot biomass. Conversely, the points located below the line ($RR = 0$) meant $RR (R/S) < 0$, indicating that warmer conditions increased the accumulation of shoot biomass more than root biomass. In the revised version, we added the description in the legend of Figure 4 (L161-164): “The points above the line of $RR(R/S) = 0$ indicate that warming condition enhanced root biomass accumulation relative to shoot biomass (i.e., $RR (R/S) > 0$), and vice versa”.

L94-97: “On average, warming significantly increased root: shoot ratio (R/S) by 6.7% with the confidence interval (CI) of intercept of 0.044-0.107 (Fig. 1e), which was

significantly larger than 0 ($p < 0.05$), agreeing with the assumption of more biomass allocation to roots in vertical dimension in warmed environments.” This sentence is a mess. First, there is the repetition – “significantly increase”, “CI was significantly larger than 0 ($p < 0.05$).” Then this is stated as “agreeing with the assumption.” However, your assumption is actually a hypothesis. Clean it up.

[Response] Good point! In the revised version, we rephrased these sentences as (L117-119): “Based on 94 pairs of observations of R/S, an average warming of 2.5 °C increased root: shoot ratio (R/S) by 6.1% - 8.8% (with a mean increase of 8.1%), which was significantly larger than 0 ($p < 0.01$, Fig. 1f). The intercept between R/S at warmed temperature and that at ambient temperature ranged from 0.044 to 0.107 (CI > 0 , $p < 0.05$, Fig. 1e), suggesting more biomass allocation to roots in vertical dimension in warmed environments (Fig. 1a). Meanwhile, the slope between R/S at warmed temperature and that at ambient temperature ranged from 0.908 to 0.997 (CI < 1 , $p < 0.05$, Fig. 1e), indicating horizontal homogenization of R/S under warmer condition (Fig. 1e)”

Responses to Reviewer # 2

The topic of this manuscript is very relevant to Nature Communications in that the study reviews the global literature to assess how warming under climate change may impacts root-to-shoot ration of different types of plant. It is recommended that the manuscript is suitable for publication after some major revisions as outline below:

[Response] We are grateful for the positive remarks on our manuscript by the reviewer. We have revised the manuscript following several constructive and helpful suggestions made by the reviewer.

Methodology: From the Abstract also the brief description of methodology (e.g. L88), it is not clear: (a) what sort of studies were collated in the review. It is not clear from reading the main report what sorts of experiments were collated. Were these glass house studies, or field studies, or both? (b) What sort of plants were included in the review. This also is not clear when reading the first part of the 'Data collection' section of the extended Methods section. What plant functional types were reviewed is not made clear until the plant functional types categories assess were introduced in L32-33 of 'Data collection' section of the extended Methods.

[Response] We agree with the reviewer that the information was not clear in our original submission. In the revised version, we added many missing information particularly to the Method section, such as the type of warming experiments, warming methods, description of different biomes, plant functional types, and plant life forms. For example, we added the following details to the Method document (L22-25): “Using the five above selection criteria, 322 papers associated with field or laboratory warming experiments were selected (Text S1 and Table S4). The methods of air and/or soil warming in selected studies included open top chamber, infrared heater, soil heating cable, and greenhouse (Table S6)”; and further (L45-51):, “The biome types (including cropland, desert, forest, grassland, tundra, and wetland, Table S5), plant functional types (PFTs, including woody vs herb, tree vs shrub, grass vs forb, Fig. S1), other taxonomy categories (evergreen vs. deciduous tree, broad vs. coniferous leaf tree, annual vs. perennial plant, angiosperm vs. gymnosperm, monoculture and mixed, Figs. 2 and 3, Table S4) and mycorrhizal fungi types (MFT, Fig. S1, Table S5) for dominant plants in each case study were confirmed by the original publications and latest FungalRoot database¹⁷”. In addition, we added details on how root or belowground biomass was measured across studies (L32-33, Method document): “The methods for roots or belowground biomass included direct harvest, soil core, and ingrowth mesh bags (Table S6)”. The potential influence from experimental methods, and the category of biomes and plants were examined using analysis of variance (ANOVA) or the Q-statistic, and the correlated results were listed in Table S4, S5, and S6 in the revised Supplementary Information.

Objectives: L81-2. This objective has not been as well introduced in the previous paragraphs. Why would we expect warning to homogenize R: across the biomes? The link between this 2nd hypothesis and the arbuscular mycorrhizal fungi and

ectomycorrhizal fungi is not clearly made.

[Response] Indeed, this link could have been clearer. We have revised the description about the possible homogenization (or differentiation) of horizontal distribution of R/S among different biomes related with mycorrhizal types as (L69-72): “Studies have shown that the dominance of a given mycorrhizal fungal type (MFT) vary in different terrestrial biomes, which associate with variable biomass allocations reported in various biomes^{18,19}. Consequently, warming effects on R/S across terrestrial biomes could depend on biome-specific MFT dominance”. We hope that the above changes help clarify the links between homogenization of R/S across biomes in response to warming and association with mycorrhizal types. Moreover, our revised figures (e.g. revised Fig. 1) also help understand these ideas better after incorporating suggestions from the reviewer 1.

L116-117. First time this part of the methodology is introduced. Suggest making assessing the attribution for change in R:S with warming a 3rd objective. For example, how were the results presented in Fig. 2 obtained? There is no description of this included.

[Response] We added the potential influence of mean annual precipitation on response of R/S as the third hypothesis in our study as (L87-89): “Third, the mean annual precipitation (MAP) would be a key factor determining the response of R/S due to direct dependency of belowground resources, such as nutrients and soil water”.

Results and Discussion: L230-237. These statements here in the Conclusions, and in the Discussion, are too strong given the variability in the collated data which is clear in Fig. 1 and 3. For example, in Fig. 3, the differences between AMF and EMF do not seem significantly different, although each could have a significant fitted relationship fitted.

[Response] We thank the reviewer for this suggestion for toning down some of our conclusions. To this, we added some of the main study limitations to the revised version as (L309-314): “Although we found some clear signal in how warming effects were observed through a greater allocation of root biomass in terrestrial plants relative to shoot biomass, we caution readers about the variability in effect sizes, which, we assume, could be related to variation in the magnitude of warming across experiments. In addition, future studies are required to understand the implication of such shifts on terrestrial carbon balance”.

L94. Unclear how the significant increase in R:S of 6.7% is calculated from Fig. 1e.

[Response] In the revised version, we added the frequency of individual response ratio of R/S among 94 pairs of observations as a new panel Fig. 1f. The p value of normal distribution test of RR (R/S) were 0.3818 (see Text S2 in Supporting information). The recalculated weighted response ratio of R/S using fixed effects model (0.078 ± 0.006 , mean \pm se) was used to replace the 6.7% (using random effects model previously) in this revision. We updated the revised result as (in L117-119): “Based on 94 pairs of observations of R/S, an average

warming of 2.5 °C increased root: shoot ratio (R/S) by 6.1% - 8.8% (with a mean increase of 8.1%), which was significantly larger than 0 ($p < 0.01$, Fig. 1f)".

Structure, or layout of the article: Suggest refining objectives as discussed above. Then arrange the article to report on these specific objectives. For example, -L170. The discussion of SOC implications in the first paragraph of the Discussion seems misplaced. Why not start with the findings in relation to your key objective first, then discuss other possible feedback effects after that? -L180. Ditto for soil nutrient effects. This seems very speculative, and not directly related to the key objectives anyway.

[Response] We thank the reviewer for suggesting these changes. We have made a number of changes to improve the structure and thereby the accessibility of our manuscript. We did this in almost all the sections of the revised manuscript. Further, we made sure to tone down our conclusions whenever the variation in results were high (e.g., L309-314). Finally, we took the suggestion of the reviewer regarding the presentation of results in the discussion section (L230-244): "Our global synthesis show that climate warming enhances plant's biomass allocation to belowground but mainly so in drier habitats, agreeing with the hypothesis of warming-induced vertical downward biomass allocation (Fig. 1)²⁰. The contingency of warming effects on plant biomass via soil moisture (Fig. S5) might be one of the primary reasons for the upregulation of R/S in many study sites, except for tundra and alpine biomes where soils are often frozen^{13,21}. Even in the studies with warming-induced decrease in total plant biomass, biomass allocations were still higher belowground relative to aboveground (Figs. S6 and S7)^{22,23}. Such a plant strategy can have important implications for soil carbon dynamics, due to the lower turnover rate of root-associated carbon (i.e., carbon in hyphae and root per se) in comparison with that of shoot^{23,24}. Biomes with lower original R/S at ambient conditions showed greater allocation to belowground biomass when warmed most likely to meet warming-induced increase in demand for resources²⁵, agreeing with our hypothesis on horizontal homogenization of R/S across biomes in a warmer world (Fig. 1). We suspect that the upregulated R/S due to warming might alter the carbon transfer between atmosphere and soil^{26,27}".

Minor comments. Fig. 2. Need to state what soil layers (or depth) is being considered in the SOC and BD variables, e.g. 0-30 cm, or 0-50 cm etc? Also, were the MAP and MAT data collated from the same global data source? How accurate were these estimates?

[Response] Done as suggested. We added the soil depth (topsoil, 0-30 cm), and soil properties, including soil organic carbon (SOC), soil bulk density (BD) and soil clay content (CLAY), used in this study in both Fig. 3's legend (in L151-156). The CLAY, BD and SOC were from the topsoil at 0-30 cm soil depth, the detailed information for these predictors is listed in Table S1 in the revised Supplementary Information (Table RT1).

Moreover, in the Method section, we added the detailed information about the source and the ranges of climate and soil properties (L33-44 in Method document): "In order to analyze the effects of climate (only field studies) and treatment variable, we extracted MAP, MAT, DUR, WM, and the soil properties (0-30 cm depth), i.e., bulk density (BD), clay content

(CLAY), and soil organic carbon (SOC) in each sites from Global Gridded Surfaces of Selected Soil Characteristics (BD from IGBP-DIS)²⁸ and the Harmonized World Soil Database (CLAY and SOC, from version 1.2, <https://daac.ornl.gov/SOILS/guides/HWSD.html>) to analyze their effect on response of concerned variables (Tables S3). The sites of the selected studies mainly distributed in East Asia, North America and Europe (Fig. S1). The MAP and MAT of sites in field experiments ranged from 27 mm to 2400 mm (median 636.2 mm), and -20 °C to 30 °C (median 7.4 °C), respectively. The warming duration and magnitude ranged from one growing season to 25 years (median 2 years), and 0.26 °C to 12 °C (median 2 °C), respectively (Table S1 and Fig. S1)".

Fig. 3. What are the grey shaded areas? Please include description in the figure caption.

[Response] The grey shaded areas represented the 95% confidence intervals. We clarified it in the legend of revised figure 4 (L163-164).

Responses to Reviewer #3

This manuscript describes results from a meta-analysis of over 300 studies showing that warming significantly increases biomass allocation to roots (13%), and that mean annual precipitation of the site, and the type of mycorrhizal fungi were mostly related to this response. In particular, this response was greater in drier than wetter sites and was more significant for plants associated with arbuscular mycorrhizal (AM) fungi than those with ectomycorrhizal (EM) fungi. These results are interesting, but not of special novelty.

[Response] We thank the reviewer for insightful remarks on our manuscript and have tried our best to address the main concerns raised by the reviewer.

The first major finding was that “climate warming enhances belowground biomass allocation in plants but mainly so in relatively drier habitats”. This finding is solid, but is largely expected. Warming often increases evapotranspiration and water stress on plants, and plants respond to this change by allocating more photosynthates belowground to exploit more water, particularly in drier habitats.

[Response] In general, we agree with the reviewer that plants should allocate more carbon belowground in the warmer environments. What is interesting and arguably a novel finding in our study is that warming-induced greater belowground biomass allocation is independent of warming effects on total plant biomass. Indeed, when warming increases total biomass, it can be assumed that it is via the increase in demand for the belowground resources. By contrast, when warming negatively affects total biomass, one could expect that it is via an overall reduced fitness in plants, and both shoot and root biomass should proportionally decrease. The latter obviously was not supported by our meta-analysis. To illustrate this point, we added a new estimation of weighted response ratios of total biomass (TB), above- and belowground biomass (AGB and BGB), and root: shoot (R/S) to warming based on studies with decreased (Figure RF6 a) and increased TB (Figure RF6 b), respectively, in Figure S6. The correlated results were added in L235-237 in this revision. Based on this, we can further confirm that warming was enhancing the biomass allocation to root relative to shoot even when the total biomass reduced in warmed environments.

Figure RF6 (Figure S6 in revised manuscript): Weighted response ratios (RR_{++}) of TB, above- and belowground biomass (AGB and BGB), and root: shoot (R/S) to warming based on studies with decreased (a) and increased (b)TB respectively.

The second major finding was that there was a difference in plant biomass allocation responses between plants with arbuscular mycorrhizal fungi and those with ectomycorrhizal fungi. This seems to be a little more interesting, but the discussion of the potential underlying mechanisms was largely speculative and highly questionable. The manuscript attempts to attribute this difference to different nutrient acquisition strategies between the two types of mycorrhizae, but ignores the fundamental differences in their host plants, particularly in terms of root systems and leaf structure. Host plants for ecto-mycorrhizal (ETM-) fungi are largely perennials and dominantly coniferous and woody plants, which are with deep root systems and often with high water use efficiency (e.g., water conserving needles with a thick, waxy coating). These plants are likely less sensitive to changes in water availability induced by warming. In contrast, many plant species associated with arbuscular mycorrhizal fungi are annuals and deciduous, including the vast majority of herbaceous grasses and forbs with shallow root systems. The shallow root system makes many AMF-associated host plants more sensitive to warming-induced moisture changes in surface soils. In this sense, the coarse comparison between these two types of plants becomes less meaningful because it does not provide sufficient information for readers to infer whether the difference observed stems from different mycorrhizae-types or different hosts.

[Response] We thank the reviewer for raising these critical points. As obligate root symbionts, most mycorrhizal fungi entirely depend on the host plant for C acquisition, which subsequently affects the carbon economics of plants, and thus are tightly linked with a number of plant traits and subsequent strategies for their resource acquisition²⁹. Moreover, a plant species also often exhibits a stable type of mycorrhizal symbiont-either arbuscular or ectomycorrhizal and rarely the both. Our meta-analysis is based on mycorrhizal associations with 164 vascular plant species. Based on the reviewer's comment that 'between-plant'

variation within mycorrhizal fungal types (MFT), and particularly so within plants associated with arbuscular mycorrhizal fungi, we supplemented our analysis by adding plant phylogeny of 164 plants (L131-134) along with other variables (Figure RF7 below, and Fig. 2 in the revised version).

Figure RF7 (Figure 2 in the revised manuscript): Phylogenetic tree of 164 vascular plant species (consisting of both gymnosperms and angiosperms) included for the estimation of weighted response ratio of root: shoot ratio (R/S), total biomass, above-, and below-ground biomass to warming, shown with their mycorrhizal association (AMF or EMF), life form (woody or herb) and growth form (annual or perennial).

More specifically, to confirm the importance of MFT on the estimation of R/S's response under warming condition, we supplemented a series analyses, including a nested analysis (Figure RF8b, as Fig. 3 in main text), heterogeneity (Q_b) of response ratio (RR) of R/S between studies conducted in field vs laboratory, evergreen vs. deciduous tree, broad vs. coniferous leaf tree, annual vs. perennial plant, angiosperm (Ang.) vs. gymnosperm (Gym.) and monoculture vs. mixed plant community (Table RT3, as Table S4 in main text, and L165-169), and finally the calculation of Blomberg's K values to test phylogenetic signal in warming response of four main response variables in our meta-analysis (Table RT2, as Table S2 in main text). Although there was a significant interaction between MFT and biomes (Table RT4, as Table S5 in main text), MFT was still the second most important factor for explaining the variation in warming effects on plant's R/S (as in the previous version), explaining a greater proportion of variance for RR (R/S) relative to types of biomes (in L170-172). The responses of R/S and TB to warming across all the vascular plant species displayed a significant phylogenetic signal (R/S: $K > 1$ and TB: $K < 1$), indicating that the

variation of warming response of R/S tends to be relatively greater among clades and not so for TB (L135-136). Although the response of R/S to warming across all vascular plant species in our meta-analysis displayed a significant phylogenetic signal, mean annual precipitation (MAP) remained the most important factor in determining the warming effects on R/S relative to all other predictors (in L135-141), which was true also in our previous analysis. In addition, when we ignored the mycorrhizal fungal types, there was no significant difference in warming response of concerned variables between evergreen vs. deciduous tree, broad vs. coniferous leaf tree, annual vs. perennial plant, or even between angiosperms and gymnosperms (L166-169, Table RT3, as Table S4 in revised manuscript). The necessary description about the differences between subgroups of plants, experimental methods and analysis are further added to L167-173 and 193-197 in revised main text and L81-84 in revised Method document. These analyses further confirm the importance of MFT on warming effects on R/S compared to many other variables, which is one of the main messages of our manuscript. We thank the reviewer for raising these concerns that now helped us to further assure the importance of MFT in determining plant's responses to warming in a changing world by incorporating several other important factors (e.g., plant phylogeny).

Figure RF8 (Figure 3 in the revised manuscript): Model-averaged importance of various predictor variables for warming effects on root: shoot ratio [RR (R/S)] (a); and the proportion of total variance of RR (R/S) explained by mycorrhizal fungal types (b). The variables with importance value >0.8 were considered as essential predictors. The importance value of predictor is based on the sum of Akaike weights derived from model selection using corrected Akaike's information criteria. MFT, mycorrhizal fungal types; PFT, plant functional types; MAP, mean annual precipitation; MAT, mean annual temperature; LAT, latitude; CLAY, the proportion of clay in soil; BD, bulk density; SOC, soil organic carbon; WM, warming magnitude; DUR, warming duration (a). The CLAY, BD and SOC were from the topsoil at 0-30 cm soil depth, the detailed information for these predictors were listed in Table S3. The types of biomes including forest, grassland, cropland, tundra and wetland; AMF, EMF, and AM-EMF indicated the biomes with dominant root symbiosis of arbuscular mycorrhizal, ecto-mycorrhizal fungi, and mixed root symbiosis of mycorrhizal fungi, respectively (b).

Table RT2 (Table S2 in the revised manuscript): Blomberg's K values for warming response ratio (RR) of root: shoot (R/S), total biomass (TB), above-, and below-ground biomass (AGB and BGB) of 164 vascular plant species in this study.

Variables	Blomberg's K	P
RR (R/S)	1.713	0.001
RR (TB)	0.104	0.024

RR (AGB)	0.065	0.479
RR (BGB)	0.031	0.848

Bold value ($P < 0.05$, or $P < 0.01$) denotes the warming response ratio of root: shoot and total biomass exhibited a significant phylogenetic signal.

Table RT3 (Table S4 in the revised manuscript): Response ratio heterogeneity (Q_b) of (*RR*) of root: shoot (R/S), total biomass (TB), above-, and below-ground biomass (AGB and BGB) between studies conducted in field vs. laboratory, evergreen vs. deciduous tree, broad vs. coniferous leaf tree, annual vs. perennial plant, angiosperm (Ang.) vs. gymnosperm (Gym.) and monoculture vs. mixed community.

Variables	Group 1	Group 2	Q_b	Sig.
RR (R/S)	Field	Laboratory	40.152	0.422
RR (R/S)	Evergreen	Deciduous	17.393	0.494
RR (R/S)	Broad	Coniferous	44.032	0.146
RR (R/S)	Annual	Perennial	273.064	0.075
RR (R/S)	Ang.	Gym.	57.926	0.398
RR (R/S)	Monoculture	Mixed	35.914	0.538
RR (TB)	Field	Laboratory	38.211	0.555
RR (TB)	Evergreen	Deciduous	25.901	0.407
RR (TB)	Broad	Coniferous	25.827	0.457
RR (TB)	Annual	Perennial	45.078	0.268
RR (TB)	Ang.	Gym.	21.865	0.440
RR (TB)	Monoculture	Mixed	9.014	0.634
RR (AGB)	Field	Laboratory	11.526	0.457
RR (AGB)	Evergreen	Deciduous	193.949	0.058
RR (AGB)	Broad	Coniferous	55.779	0.470
RR (AGB)	Annual	Perennial	343.031	0.224
RR (AGB)	Ang.	Gym.	31.901	0.633
RR (AGB)	Monoculture	Mixed	425.553	0.147
RR (BGB)	Field	Laboratory	25.457	0.573
RR (BGB)	Evergreen	Deciduous	0.724	0.894
RR (BGB)	Broad	Coniferous	44.524	0.333
RR (BGB)	Annual	Perennial	109.976	0.112
RR (BGB)	Ang.	Gym.	138.519	0.075
RR (BGB)	Monoculture	Mixed	21.417	0.486

* indicated the significant difference at $p < 0.05$.

Table RT4 (Table S5 in revised manuscript): The results of ANOVA to test the effects of biomes (including cropland, desert, forest, grassland, tundra, and wetland), plant functional types (PFT, including grass, forb, shrub and tree) and mycorrhizal fungi types (MFT, including AM, EM, and AM-EMF) on responses of plant total biomass (TB), aboveground biomass (AGB), belowground biomass (BGB) and root: shoot (R/S) to warming.

Sources	Variables	df	MS	F	Sig.
Biome	RR (TB)	4	0.271	1.088	0.366
PFT	RR (TB)	3	0.526	2.113	0.102
MFT	RR (TB)	2	0.520	2.089	0.129
Biome×PFT	RR (TB)	2	0.436	1.754	0.178

Biome×MFT	RR (TB)	4	0.332	1.334	0.262
PFT×MFT	RR (TB)	0	--	--	--
Biome×PFT×MFT	RR (TB)	0	--	--	--
Biome	RR (AGB)	4	0.200	1.863	0.118
PFT	RR (AGB)	3	0.152	1.119	0.244
MFT	RR (AGB)	2	0.582	5.421	0.001**
Biome×PFT	RR (AGB)	4	0.112	1.044	0.385
Biome×MFT	RR (AGB)	2	0.029	0.271	0.763
PFT×MFT	RR (AGB)	1	0.095	0.888	0.347
Biome×PFT×MFT	RR (AGB)	0	--	--	--
Biome	RR (BGB)	5	0.547	5.282	0.000***
PFT	RR (BGB)	3	0.645	6.226	0.001**
MFT	RR (BGB)	2	0.216	2.090	0.129
Biome×PFT	RR (BGB)	3	0.075	0.724	0.540
Biome×MFT	RR (BGB)	1	0.301	2.905	0.091
PFT×MFT	RR (BGB)	1	0.000	0.001	0.978
Biome×PFT×MFT	RR (BGB)	0	--	--	--
Biome	RR (R/S)	4	0.065	0.743	0.566
PFT	RR (R/S)	3	0.187	2.131	0.104
MFT	RR (R/S)	2	0.124	1.417	0.249
Biome×PFT	RR (R/S)	2	0.145	1.647	0.200
Biome×MFT	RR (R/S)	1	0.370	4.215	0.044*
PFT×MFT	RR (R/S)	0	--	--	--
Biome×PFT×MFT	RR (R/S)	0	--	--	--

*, $p < 0.05$; **, $p < 0.01$; ***, $p < 0.001$

References

- 1 Henneron, L., Cros, C., Picon-Cochard, C., Rahimian, V. & Fontaine, S. Plant economic strategies of grassland species control soil carbon dynamics through rhizodeposition. *Journal of Ecology* **108**, 528-545, doi:10.1111/1365-2745.13276 (2020).
- 2 Arft, A. M. *et al.* Responses of tundra plants to experimental warming: Meta-analysis of the international tundra experiment. *Ecological Monographs* **69**, 491-511, doi:10.1890/0012-9615(1999)069[0491:Rotpte]2.0.Co;2 (1999).
- 3 Bloom, A. A., Exbrayat, J.-F., van der Velde, I. R., Feng, L. & Williams, M. The decadal state of the terrestrial carbon cycle: Global retrievals of terrestrial carbon allocation, pools, and residence times. *Proceedings of the National Academy of Sciences of the United States of America* **113**, 1285-1290, doi:10.1073/pnas.1515160113 (2016).
- 4 Ma, H. *et al.* The global distribution and environmental drivers of aboveground versus belowground plant biomass. *Nature Ecology & Evolution* **5**, 1110-+, doi:10.1038/s41559-021-01485-1 (2021).
- 5 Mokany, K., Raison, R. J. & Prokushkin, A. S. Critical analysis of root: shoot ratios in terrestrial biomes. *Global Change Biol.* **12**, 84-96, doi:10.1111/j.1365-2486.2005.001043.x (2006).

- 6 Shipley, B. & Meziane, D. The balanced-growth hypothesis and the allometry of leaf and root biomass allocation. *Functional Ecology* **16**, 326-331, doi:org/10.1046/j.1365-2435.2002.00626.x (2002).
- 7 Piao, S. *et al.* Net carbon dioxide losses of northern ecosystems in response to autumn warming. *Nature* **451**, 49-52, doi:10.1038/nature06444 (2008).
- 8 Kim, J.-S. *et al.* Reduced North American terrestrial primary productivity linked to anomalous Arctic warming. *Nature Geosci.* **10**, 572-576, doi:10.1038/ngeo2986 (2017).
- 9 Lin, D., Xia, J. & Wan, S. Climate warming and biomass accumulation of terrestrial plants: a meta- analysis. *New Phytol.* **188**, 187-198, doi:org/10.1111/j.1469-8137.2010.03347.x (2010).
- 10 Fernandez, C. W. *et al.* Ectomycorrhizal fungal response to warming is linked to poor host performance at the boreal-temperate ecotone. *Global Change Biol.* **23**, 1598-1609, doi:10.1111/gcb.13510 (2017).
- 11 Keller, J. A. & Shea, K. Warming and shifting phenology accelerate an invasive plant life cycle. *Ecology* **102**, e03219, doi:10.1002/ecy.3219 (2020).
- 12 Rasheed, M. U. *et al.* The responses of shoot-root-rhizosphere continuum to simultaneous fertilizer addition, warming, ozone and herbivory in young Scots pine seedlings in a high latitude field experiment. *Soil Biol. Biochem.* **114**, 279-294, doi:10.1016/j.soilbio.2017.07.024 (2017).
- 13 Xu, M., Liu, M., Xue, X. & Zhai, D. Warming effects on plant biomass allocation and correlations with the soil environment in an alpine meadow, China. *Journal of Arid Land* **8**, 773-786, doi:10.1007/s40333-016-0013-z (2016).
- 14 Averill, C., Dietze, M. C. & Bhatnagar, J. M. Continental-scale nitrogen pollution is shifting forest mycorrhizal associations and soil carbon stocks. *Global Change Biol.* **24**, 4544-4553, doi:10.1111/gcb.14368 (2018).
- 15 Steidinger, B. S. *et al.* Climatic controls of decomposition drive the global biogeography of forest-tree symbioses. *Nature* **569**, 404-408, doi:10.1038/s41586-019-1128-0 (2019).
- 16 IPCC. Summary for Policymakers. In: *Climate Change 2021: The Physical Science Basis. Contribution of Working Group I to the Sixth Assessment Report of the Intergovernmental Panel on Climate Change.* 1535 (Cambridge University Press, 2021).
- 17 Soudzilovskaia, N. A. *et al.* Global mycorrhizal plant distribution linked to terrestrial carbon stocks. *Nature Communications* **10**, 5077, doi:10.1038/s41467-019-13019-2 (2019).
- 18 Hollister, R. D. & Flaherty, K. J. Above- and below-ground plant biomass response to experimental warming in northern Alaska. *Applied Vegetation Science* **13**, 378-387, doi:10.1111/j.1654-109X.2010.01079.x (2010).
- 19 Johnson, N. C., Rowland, D. L., Corkidi, L. & Allen, E. B. Plant winners and losers during grassland N-eutrophication differ in biomass allocation and mycorrhizas. *Ecology* **89**, 2868-2878, doi:10.1890/07-1394.1 (2008).
- 20 Litton, C. M. & Giardina, C. P. Below- ground carbon flux and partitioning: global patterns and response to temperature. *Functional Ecology* **22**, 941-954 (2008).
- 21 Wang, P. *et al.* Belowground plant biomass allocation in tundra ecosystems and its relationship with temperature. *Environmental Research Letters* **11**, 055003 (2016).
- 22 Hovenden, M. J. *et al.* Warming and elevated CO₂ affect the relationship between seed

- mass, germinability and seedling growth in *Austrodanthonia caespitosa*, a dominant Australian grass. *Global Change Biology* **14**, 1633-1641 (2008).
- 23 Olszyk, D. M. *et al.* Whole-seedling biomass allocation, leaf area, and tissue chemistry for Douglas-fir exposed to elevated CO₂ and temperature for 4 years. *Canadian Journal of Forest Research* **33**, 269-278 (2003).
 - 24 Hovenden, M. J. *et al.* Warming and elevated CO₂ affect the relationship between seed mass, germinability and seedling growth in *Austrodanthonia caespitosa*, a dominant Australian grass. *Global Change Biol.* **14**, 1633-1641, doi:10.1111/j.1365-2486.2008.01597.x (2008).
 - 25 Parmesan, C. & Hanley, M. E. Plants and climate change: complexities and surprises. *Annals of Botany* **116**, 849-864 (2015).
 - 26 Poorter, H. *et al.* Biomass allocation to leaves, stems and roots: meta-analyses of interspecific variation and environmental control. *New Phytol.* **193**, 30-50, doi:10.1111/j.1469-8137.2011.03952.x (2012).
 - 27 Hagedorn, F., Gavazov, K. & Alexander, J. M. Above- and belowground linkages shape responses of mountain vegetation to climate change. *Science* **365**, 1119-+, doi:10.1126/science.aax4737 (2019).
 - 28 Task Group, G. S. D. (Oak Ridge National Laboratory Distributed Active Archive Center, Oak Ridge, Tennessee, U.S.A. , 2000).
 - 29 Bergmann, J. *et al.* The fungal collaboration gradient dominates the root economics space in plants. *Science Advances* **6**, doi:10.1126/sciadv.aba3756 (2020).

Reviewers' Comments:

Reviewer #1:

Remarks to the Author:

I have read through the revised manuscript and the author responses to my comments on the previous manuscript (in addition to the other referees). The authors have seriously engaged with each critique.

I have a few remaining issues, however, which they might address.

Magnitude of warming

It is useful to have the magnitude of warming listed as meta-data... But why not include it as a covariate in the analysis? It strikes me as a potentially confounding factor in nearly every analysis. I would like to see either that it could be removed via model-selection (where it is not contributing) or to explore the main and interactive effects of warming magnitude.

Figure 1.

IN the previous manuscript, I described how it was difficult to visualize an effect in Figure 1 (a, upward vs. b, downward integration; c, homogenization vs. d, differentiation). I suggested that the effect, as shown as the difference between the blue dotted 1:1 line and the best-fit line, might appear small because of the large, inter-experimental variability in R/S ratios. (e.g., the scale of the x-axis goes from -2 to 1, whereas there is no residual of greater than 0.5.

Thus, what I hoped to see in revision is a plot with $\log(R/S \text{ under ambient})$ on the x-axis and the residual $\log(R/S \text{ under warming})$ on the y. In keeping with the statistical hypothesis testing laid out in Figure 1abcd, homogenization vs. differentiation would now be testing for differences from a (null) slope of 0.

In revision, the authors provided a new panel (g) which shows R/S under ambient condition on the x-axis (not log transformed) vs. residuals from the 1:1 line on the y-axis.

Below I made a figure of my own, using the following simulated data in R:

```
# make dummy data, n=100
x=runif(100, min=-1, max=2)
# give a positive intercept of .1 and a slope less than 1 (.9)
y=rnorm(100,.1,.1)+rnorm(100,.9,.1)*x

# look at the summary stats
summary(lm(y~x))
summary(lm((y-x)~x))

# plot
plot(x,y,main="Figure 1a",xlab="", ylab="log(r/s warmed)")
abline(a=0, b=1, col="blue",lty=2, lwd=2)
abline(lm(y~x), col=2, lwd=2)

plot(x, y-x, main="this is what I wanted", xlab="log(r/s ambient)", ylab=" log(r/s warmed)-
log(r/s ambient)")
abline(h=0,col="blue",lty=2, lwd=2 )
```

```
abline(lm((y-x)~x), col=2, lwd=2)
```

You can see that in the bottom figure the difference between the line and the 0-intercept is clearer. As is the slope effect. Further, the uncanny fit ($R^2 = 0.96$ in the top), which you get from plotting ambient vs. warming condition against one another in Figure 1a, is reduced in the bottom figure ($R^2 = 0.27$ on bottom). You are now looking at the effect of ambient

$\log(r/s)$ on the difference in allocation ratios due to warming. I feel this more directly addresses your hypothesis.

L60-61. Plants with the same mycorrhizal association generally respond in (ecologically) similar ways to global change factors, especially to alteration of resource availability in soils^{24,25}

I appreciate you added the reference I suggested. I would stress, however, that whereas the Averill paper shows the impact of atmospheric N-deposition on tree-mycorrhizal composition, the Steidinger et al. paper focuses on climate, finding soil resources to have a comparatively minor influence. Thus they should not be cited together in this way.

For Figure 6 (the Path Analysis) – I am curious if, in keeping with Reviewer 3 concerns, it is necessary to have an arrow from Plant Functional Type to Mycorrhizal Functional Type, as the only ectomycorrhizal taxa will be trees.

Reviewer #2:

Remarks to the Author:

The revised manuscript has been significantly improved in response to reviewers comments. The revised manuscript is now suitable for publication.

Reviewer #3:

Remarks to the Author:

This new version has significantly improved. The analysis of the relationship between plant phylogeny and warming effect is also helpful. Yet, it did not fully address my major concern, that is, "do mycorrhizal types REGULATE or DRIVE the warming effect on plant biomass allocation patterns"?

To claim that different mycorrhizal types (mainly arbuscular mycorrhizae, AM and ecto-mycorrhizae, EcM in this manuscript) regulate or drive the warming effect on plant biomass allocation patterns, it should present some evidence showing that

1. arbuscular mycorrhizal fungi have significantly different effects on plant biomass allocation under warming from ecto-mycorrhizal fungi have. This means evidence from experiments that manipulated mycorrhizal fungi (e.g., elimination or addition) under similar warming conditions,

2. for tree plant species that are associated with both arbuscular and ecto-mycorrhizal fungi, elimination of one type or another leads to significantly different responses of plant biomass allocation to warming, or

3. at least, the plant biomass allocation pattern of AM-plants responded to warming significantly different from that of EcM plants under the same warming conditions (this might occur on mixed forest experiments, but do we have evidence?). Has any study shown this difference or no difference? The discussion should include these results, if any.

Yes, the dataset showed that there was a difference between two types of mycorrhizal plants, but the statement "mycorrhizal associations regulate or drive...." is way beyond the conclusion the dataset can substantiate. The current dataset only showed a correlation relationship, but not a cause-effect one.

Therefore, I recommend modifying the wording across the manuscript (the title, summary, discussions) to better reflect what the dataset and analyses can substantiate.

Responses to Reviewer #1

I have read through the revised manuscript and the author responses to my comments on the previous manuscript (in addition to the other referees). The authors have seriously engaged with each critique.

[Response] We thank the reviewer for the positive and constructive comments, which have helped us to improve our manuscript.

I have a few remaining issues, however, which they might address. Magnitude of warming: It is useful to have the magnitude of warming listed as meta-data, But why not include it as a covariate in the analysis? It strikes me as a potentially confounding factor in nearly every analysis. I would like to see either that it could be removed via model-selection (where it is not contributing) or to explore the main and interactive effects of warming magnitude.

[Response] We agree with the reviewer that the warming magnitude (WM) (and also the warming duration, DUR) could potentially explain the variability in the effect size of biomass allocation across warming studies. In order to analyze how WM and DUR influence the effect sizes, we first categorized WM and DUR into six classes, respectively (Table R1, based on Lin *et al.* (2010)). Across all studies, we didn't find any significant main or interactive effects of WM and DUR on the variability of warming-induced effects on plant biomass or biomass allocation (Figure R1, Table R2, as Table S6 in the main manuscript). Although when we ran stepwise regression models with several variables including WM and DUR for specific plant and mycorrhizal groups, WM displayed a negative relationship with *RR* (*R/S*) in EMF groups, while DUR had a negative relationship with *RR* (*R/S*) in woody but a positive relationship in herbaceous plants (Table R3, as Table S3 in the main manuscript). Therefore, we incorporated WM and DUR into the analysis of importance of various predictor variables for warming effects on root: shoot ratio (Figure R2), and also in structural equation model (Figure R3).

Based on these new analyses, we found that WM and DUR exhibited no significant contribution in explaining the variation in *RR* (*R/S*) across different warming experiments. In the revised version, we added a sentence to address these results (Lines 175-177): “Moreover, both warming magnitude and warming duration didn't affect the *RR* (*R/S*) across warming experiments used in our meta-analysis (Table S6, Fig. 3a)”. However, given that we found WM and DUR effects in our stepwise regression with multiple variables (Table R3 or Table S3 in the supplementary information of the main manuscript) we also added a sentence in the Discussion section to highlight on the importance of these two factors (Lines 309-316): “Although we found some clear signal

in how warming effects were observed through a greater allocation of root biomass in terrestrial plants relative to shoot biomass, we caution readers about the variability in effect sizes, which could be due to the distinct effect of warming magnitude and warming duration on a certain type of biomes (Table S3). For example, warming magnitude might negatively affect the response of R/S for biomes with EMF, while warming duration could induce negative and positive effect for biomes with woody and herbaceous plants, respectively as revealed in our analysis (details in Table S3)”. We again thank the reviewer for pointing this, which has further made our results clear on when warming magnitude and duration effects are important to consider in understanding of plant biomass allocation strategies in a changing world.

Table R1 Levels of subgroups for warming magnitude (WM) and warming duration (DUR) of selected studies in the meta-analysis according to Lin *et al.* (2010).

Variables	Levels of subgroups					
WM (°C)	< 1	1 - 2	2 - 3	3 - 4	4 - 5	> 5
DUR (year)	<1	1 - 2	2 - 3	3 - 4	4 - 5	> 5

Table R2 (Table S6 in Supporting Information) The results of ANOVA to test the effects of warming magnitude and duration on response ratio of root: shoot.

Sources	Variables	df	MS	F	Sig.
Warming magnitude (WM)	RR(R/S)	5	0.050	0.503	0.772
Warming duration (DUR)	RR(R/S)	5	0.059	0.597	0.702
WM × DUR	RR(R/S)	5	0.064	0.648	0.664
Warming magnitude (WM)	RR(TB)	5	0.189	0.708	0.618
Warming duration (DUR)	RR(TB)	5	0.244	0.912	0.476
WM × DUR	RR(TB)	17	0.150	0.560	0.915
Warming magnitude (WM)	RR(AGB)	5	0.132	0.873	0.500
Warming duration (DUR)	RR(AGB)	5	0.043	0.284	0.922
WM × DUR	RR(AGB)	19	0.103	0.681	0.836
Warming magnitude (WM)	RR(BGB)	5	0.052	0.356	0.877
Warming duration (DUR)	RR(BGB)	5	0.073	0.499	0.776
WM × DUR	RR(BGB)	18	0.227	1.554	0.087

The levels of subgroups of warming magnitude (WM, increased <1, 1-2, 2-3, 3-4, 4-5, >5 °C, respectively), and warming duration (DUR, for <1, 1-2, 2-3, 3-4, 4-5, >5 year, respectively) of selected studies in the meta-analysis.

Figure R1 The relationship of response ratio (RR) of root: shoot (R/S) with warming magnitude (a) and warming duration (b).

Table R3 (Table S3 in Supporting Information) Stepwise linear regression of response ratio (*RR*) of root: shoot (*R/S*) with predictor variables for biomes with different plants.

Groups	Equation	R2	P-Value
AMF	$RR (R/S) = -0.001MAP + 0.591$	0.353	<0.001
	$RR (R/S) = -0.001MAP - 0.033MAT + 0.803$	0.481	<0.001
EMF	$RR (R/S) = -0.032MAT + 0.284$	0.328	0.003
	$RR (R/S) = -0.029MAT - 0.067WM + 0.419$	0.450	0.002
Woody plant	$RR (R/S) = -0.001MAP + 0.517$	0.304	<0.001
	$RR (R/S) = -0.001MAP - 0.044DUR + 0.667$	0.408	<0.001
Herb	$RR (R/S) = 0.030DUR - 0.174$	0.171	0.003
	$RR (R/S) = 0.028DUR + 0.000MAP + 0.110$	0.312	<0.001
Tree	$RR (R/S) = 0.029DUR - 0.001MAP + 0.016MAT + 0.187$	0.386	<0.001
	$RR (R/S) = -0.001MAP + 0.508$	0.271	0.003
Shrub	$RR (R/S) = -0.001MAP + 0.008CLAY + 0.666$	0.382	0.002
	$RR (R/S) = -0.059MAT + 0.539$	0.632	0.006
Forb	$RR (R/S) = -0.077MAT - 4.884BD + 7.286$	0.966	<0.001
	$RR (R/S) = -0.001MAP + 0.178$	0.805	<0.001

RR(*R/S*), response ratio of root: shoot ratio; AMF, biomes dominated by root symbiosis of arbuscular mycorrhizal; EMF, biomes dominated by root symbiosis of ecto-mycorrhizal fungi; Clay, soil clay content (%); BD, bulk density (g/cm³); SOC, soil organic carbon (g/kg); LAT, latitude (°); MAT, mean annual temperature (°C); MAP, mean annual precipitation (mm); DUR, experimental duration (year); WM, warming magnitude (°C).

Figure R2 (Figure 3a in the main manuscript) Model-averaged importance of various predictor variables for warming effects on root: shoot ratio [*RR* (*R/S*)]. MFT, mycorrhizal fungal types; PFT, plant functional types; MAP, mean annual precipitation; MAT, mean annual temperature; LAT, latitude; CLAY, the proportion of clay in soil; BD, bulk density; SOC, soil organic carbon; WM, warming magnitude; DUR, warming duration.

Figure R3 (Figure 6 a in the main manuscript) Path analysis examining the effects of plant functional types (PFTs, e.g., herbs or woody plants), dominant mycorrhizal types of root symbiosis (MFTs, e.g., AMF, AM-EMF or EMF), background climate condition (MAT and MAP) and warming treatment (warming magnitude and duration) on response of root: shoot ratio (R/S) through changes in AGB and BGB under warming condition. The solid and dashed lines indicated the significant ($p < 0.05$) and non-significant ($p > 0.05$), respectively, whereas the red and blue solid arrows indicate positive and negative effects ($p < 0.05$), respectively. We assigned ‘0’ and ‘1’ for herbs and woody plants, respectively; ‘0’, ‘0.5’, ‘1’ to AMF, AM-EMF and EMF, respectively in this analysis.

Figure 1. In the previous manuscript, I described how it was difficult to visualize an effect in Figure 1 (a, upward vs. b, downward integration; c, homogenization vs. d, differentiation). I suggested that the effect, as shown as the difference between the blue dotted 1:1 line and the best-fit line, might appear small because of the large, inter-experimental variability in R/S ratios. (e.g., the scale of the x-axis goes from -2 to 1, whereas there is no residual of greater than 0.5. Thus, what I hoped to see in revision is a plot with $\log(R/S \text{ under ambient})$ on the x-axis and the residual $\log(R/S \text{ under warming})$ on the y. In keeping with the statistical hypothesis testing laid out in Figure 1abcd, homogenization vs. differentiation would now be testing for differences from a (null) slope of 0. In revision, the authors provided a new panel (g) which shows R/S under ambient condition on the x-axis (not log transformed) vs. residuals from the 1:1 line on the y-axis. Below I made a figure of my own, using the following simulated data in R:

```
# make dummy data, n=100
x=runif(100, min=-1, max=2)
# give a positive intercept of .1 and a slope less than 1 (.9)
y=rnorm(100,.1,.1)+rnorm(100,.9,.1)*x
# look at the summary stats
summary(lm(y~x))
summary(lm((y-x)~x))
# plot
plot(x,y,main="Figure 1a",xlab="", ylab="log(r/s warmed)")
```

```

abline(a=0, b=1, col="blue", lty=2, lwd=2)
abline(lm(y~x), col=2, lwd=2)
plot(x, y-x, main="this is what I wanted", xlab="log(r/s ambient)", ylab="log(r/s warmed)-
log(r/s ambient)")
abline(h=0,col="blue",lty=2, lwd=2 )
abline(lm((y-x)~x), col=2, lwd=2)

```

You can see that in the bottom figure the difference between the line and the 0-intercept is clearer. As is the slope effect. Further, the uncanny fit ($R^2 = 0.96$ in the top), which you get from plotting ambient vs. warming condition against one another in Figure 1a, is reduced in the bottom figure ($R^2 = 0.27$ on bottom). You are now looking at the effect of ambient $\log(r/s)$ on the difference in allocation ratios due to warming. I feel this more directly addresses your hypothesis.

[Response] We thank the reviewer for providing the clarification, and we agree with the suggested approach. Indeed, the relationship between R/S in ambient temperature condition and that under warming condition were used to test the first and second hypotheses (Lines 76-88) in our study. Based on the significant difference of intercept and slope from 0 and 1, respectively, we inferred that warming would induce greater biomass allocation downwards and homogenize R/S across the biomes (Figure R4). Although there is a significant correlation between Log (R/S under warming condition) and Log (R/S under ambient condition (Figure R4e), there was no significant linear

correlation between “Log (R/S under warming condition)-Log (R/S in ambient condition)” and Log (R/S in ambient condition) (Figure R5). Based on the reviewer’s suggestion, we now replaced the previous “Figure 1g” by a new plot (Figure R6), to show the linear correlation between “Log (residuals of R/S under warming condition from 1:1 line)” and Log (R/S in ambient condition), which might better illustrate one of our main findings (Lines125-127): “The biomes with greater R/S in ambient condition had greater variation in R/S in response to warming (Fig. 1g and S2)”.

Figure R4 (Figure 1 in the main manuscript) Hypotheses of warming effects on root: shoot ratio (R/S, a, b, c, d) and the actual relationship between log₁₀-transformed R/S with experimental warming and at ambient temperature (e). The gray shadow in panel e represents the 95% CI. Across studies used in our global synthesis, the intercept in panel e was larger than 0, implying greater biomass is allocated (vertically) downwards, i.e., greater R/S under warming than that at ambient temperature (a and e); the opposite scenario of downward allocation (i.e., upward allocation, intercept smaller than zero) is illustrated in b. When the slope is smaller than 1, it would imply that warming decreased the horizontal variability of R/S, which indicate homogenization of R/S among diverse biomes due to warming as found in our synthesis (c and e); the opposite scenario of homogenization (differentiation, slope greater than 1) is illustrated in d. The size of each dot in panel e indicate relative weight of the individual response ratio of R/S.

Figure R5 The relationship between “Log 10-transformed R/S under warming condition - Log 10-transformed R/S in ambient condition” and “Log 10-transformed R/S in ambient condition”.

Figure R6 (Figure 1g in the main manuscript) The relationship between Log 10-transformed R/S in ambient condition and Log 10-transformed residuals from the 1:1 line

L60-61. Plants with the same mycorrhizal association generally respond in (ecologically) similar ways to global change factors, especially to alteration of resource availability in soils^{24,25}. I appreciate you added the reference I suggested. I would stress, however, that whereas the Averill paper shows the impact of atmospheric N-deposition on tree-mycorrhizal composition, the Steidinger et al. paper focuses on

climate, finding soil resources to have a comparatively minor influence. Thus they should not be cited together in this way.

[Response] We thank the reviewer for pointing this out. In this revision, we kept the Averill *et al.* (2018) to support the sentence (Lines 63-64): “Plants with the same mycorrhizal association generally respond in (ecologically) similar ways to global change factors, especially to alteration of resource availability in soils”. The reference of Steidinger *et al.* 2019 was kept in the Discussion section (Lines 288-289): “By contrast, AMF-associated plants are often more abundant in warmer sites (*e.g.*, in temperate grasslands and deciduous forests) with drier soils”

For Figure 6 (the Path Analysis) – I am curious if, in keeping with Reviewer 3 concerns, it is necessary to have an arrow from Plant Functional Type to Mycorrhizal Functional Type, as the only ectomycorrhizal taxa will be trees.

[Response] We thank the reviewer for this suggestion. In the revised version, we added a new path relation between PFTs and MFTs to demonstrate their positive correlation (0.383, $P=0.002$), when we assigned ‘0’ and ‘1’ for herbs and woody plants, ‘0’, ‘0.5’, ‘1’ to AMF, AM-EMF and EMF, respectively (Figure R3, as Figure 6 in the main manuscript).

Responses to Reviewer #2

The revised manuscript has been significantly improved in response to reviewers comments. The revised manuscript is now suitable for publication.

[Response] We thank the reviewer for the positive assessment of our study.

Responses to Reviewer #3

This new version has significantly improved. The analysis of the relationship between plant phylogeny and warming effect is also helpful.

[Response] We thank the reviewer for positive assessment of our last revision and for the further comments.

Yet, it did not fully address my major concern, that is, “do mycorrhizal types REGULATE or DRIVE the warming effect on plant biomass allocation patterns”? To claim that different mycorrhizal types (mainly arbuscular mycorrhizae, AM and ecto-mycorrhizae, EcM in this manuscript) regulate or drive the warming effect on plant biomass allocation patterns, it should present some evidence showing that: 1. arbuscular mycorrhizal fungi have significantly different effects on plant biomass allocation under warming from ecto-mycorrhizal fungi have. This means evidence from experiments that manipulated mycorrhizal fungi (e.g., elimination or addition) under similar warming conditions; 2. for tree plant species that are associated with both arbuscular and ecto-mycorrhizal fungi, elimination of one type or another leads to significantly different responses of plant biomass allocation to warming; or, 3. at least, the plant biomass allocation pattern of AM-plants responded to warming significantly different from that of EcM plants under the same warming conditions (this might occur on mixed forest experiments, but do we have evidence?). Has any study shown this difference or no difference? The discussion should include these results, if any. Yes, the dataset showed that there was a difference between two types of mycorrhizal plants, but the statement “mycorrhizal associations regulate or drive....” is way beyond the conclusion the dataset can substantiate. The current dataset only showed a correlation relationship, but not a cause-effect one. Therefore, I recommend modifying the wording across the manuscript (the title, summary, discussions) to better reflect what the dataset and analyses can substantiate.

[Response] We are grateful to the reviewer for cautioning us again about the causal relationship. While many studies have shown the causal relationship between global-change factors' effects on biomass allocation and plant's association with mycorrhizal fungi (e.g., Dong et al., 2018, Hu et al., 2015, Kilpelainen et al., 2020, Kivlin et al., 2013, Wang et al., 2016), we agree that our meta-analysis cannot confirm this causal relationship. Our meta-analysis mainly points towards the importance of mycorrhizal specific association in determining plant's biomass allocation strategies to warming together with precipitation variability. We believe that this signal is strong but certainly not causative as we rarely found any warming experiments reporting root: shoot ratio with both arbuscular- (AMF) and ecto-mycorrhizal fungi (EMF)-plants, or that with manipulated AMF and EMF treatments. The reviewer is therefore spot on, and we have made sure in the revised version that we present our results more as to highlight the

importance of mycorrhiza and not exerting any causal relationship.

We also have revised several sections of our manuscript to support our findings with more plausible explanations and remain more careful about the causality. Some examples are outlined below:

Title of the study: Warming effects on biomass allocation associate with precipitation and mycorrhiza in terrestrial plants.

Abstract: Lines 29-30: “.....and two factors associate strongly with this response: mean annual precipitation of the site, and the type of mycorrhizal fungi associated with plants”, and Lines 36-38: “Our study highlights that the wetness or dryness of a site and plants’ mycorrhizal associations can strongly affect terrestrial carbon dynamics by regulating biomass allocation strategies of plants in a warmer world.”

Several sentences in the discussion, such as, Lines 287-290: “EMF associated plants showed an increasing trend of AGB, BGB and R/S under warming (Fig. 5), likely due to hypha-facilitated soil organic matter (SOM) degradation and root nutrient absorption⁴⁹, confirms this assumption” and Lines 306-309: “We suspect that stronger association of warming effects on biomass allocation with precipitation and mycorrhizal associations could potentially lead to a redistribution of R/S in horizontal dimension with more homogenization and lower variability of biomass allocation patterns among diverse biomes (Fig. 1)”.

References

- Dong Y, Wang Z, Sun H, Yang W, Xu H (2018) The Response Patterns of Arbuscular Mycorrhizal and Ectomycorrhizal Symbionts Under Elevated CO₂: A Meta-Analysis. *Frontiers in Microbiology*, **9**.
- Hu Y, Wu S, Sun Y *et al.* (2015) Arbuscular mycorrhizal symbiosis can mitigate the negative effects of night warming on physiological traits of *Medicago truncatula* L. *Mycorrhiza*, **25**, 131-142.
- Kilpelainen J, Aphalo PJ, Barbero-Lopez A, Adamczyk B, Nipu SA, Lehto T (2020) Are arbuscular-mycorrhizal *Alnus incana* seedlings more resistant to drought than ectomycorrhizal and nonmycorrhizal ones? *Tree Physiology*, **40**, 782-795.
- Kivlin SN, Emery SM, Rudgers JA (2013) Fungal symbionts alter plant responses to global change. *American Journal of Botany*, **100**, 1445-1457.
- Lin D, Xia J, Wan S (2010) Climate warming and biomass accumulation of terrestrial plants: a meta-analysis. *New Phytologist*, **188**, 187-198.
- Wang X, Peng L, Jin Z (2016) Effects of AMF inoculation on growth and photosynthetic physiological characteristics of *Sinocalycanthus chinensis* under conditions of simulated warming. *Acta Ecologica Sinica*, **36**, 5204-5214.

Reviewers' Comments:

Reviewer #1:

Remarks to the Author:

I thank the authors for their thorough response to the reviewer comments. The paper is interesting, timely, and novel. I have no additional input.

Reviewer #3:

Remarks to the Author:

This revision has addressed my concerns on the previous versions and is now acceptable for publication.

Congratulations to the authors for this comprehensive synthesis of warming effects on plant biomass allocation.